computational biology/ecology/health and disease and epidemiology

*Borrelia burgdorferi*, tick-borne disease, *Ixodes scapularis*, agent-based model, host dispersal, tick burden

**Author for correspondence:**
Olivia Tardy
e-mail: olivia.tardy@umontreal.ca

# Context-dependent host dispersal and habitat fragmentation determine heterogeneity in infected tick burdens: an agent-based modelling study

Olivia Tardy[1], Christian E. Vincenot[2], Catherine Bouchard[1,3], Nicholas H. Ogden[1,3] and Patrick A. Leighton[1]

[1]Research Group on Epidemiology of Zoonoses and Public Health, Faculty of Veterinary Medicine, Université de Montréal, 3200 rue Sicotte, Saint-Hyacinthe, Quebec, Canada J2S 2M2
[2]Department of Social Informatics, Kyoto University, Yoshida-honmachi, Sakyo-ku, Kyoto 606-8501, Japan
[3]Public Health Risk Sciences Division, National Microbiology Laboratory, Public Health Agency of Canada, 3200 rue Sicotte, Saint-Hyacinthe, Quebec, Canada J2S 2M2

OT, 0000-0002-0242-8137; NHO, 0000-0002-1062-7283

As the incidence of tick-borne diseases has sharply increased over the past decade, with serious consequences for human and animal health, there is a need to identify ecological drivers contributing to heterogeneity in tick-borne disease risk. In particular, the relative importance of animal host dispersal behaviour in its three context-dependent phases of emigration, transfer and settlement is relatively unexplored. We built a spatially explicit agent-based model to investigate how the host dispersal process, in concert with the tick and host demographic processes, habitat fragmentation and the pathogen transmission process, affects infected tick distributions among hosts. A sensitivity analysis explored the impacts of different input parameters on infected tick burdens on hosts and infected tick distributions among hosts. Our simulations indicate that ecological predictors of infected tick burdens differed among the post-egg life stages of ticks, with tick attachment and detachment, tick questing activity and pathogen transmission dynamics identified as key processes, in a coherent way. We also found that the type of host settlement strategy and the proportion of habitat suitable for hosts determined super-spreading of infected

ticks. We developed a theoretical mechanistic framework that can serve as a first step towards applied studies of on-the-ground public health intervention strategies.

# 1. Introduction

Understanding the causes of heterogeneity in macroparasite (e.g. arthropods, helminths) infestation patterns is an important and challenging research priority in disease ecology [1,2]. It is widely recognized that macroparasite burdens on their wildlife hosts are highly heterogeneous such as most hosts harbour a small number of macroparasites, while a few hosts feed the majority of macroparasites [1,3]. This observation is often in accordance with the general '20/80 rule', which states that approximately 20% of individuals in a host population are responsible for at least 80% of pathogen transmission potential [4]. Macroparasite aggregation on hosts has important implications for pathogen transmission and spread given that the basic reproduction number, $R_0$, which is an indicator of the success of pathogen invasion into a naive population [5–7], increases with the degree of macroparasite aggregation [4]. It is thus crucial to identify ecological drivers that contribute to heterogeneity in macroparasite burdens among hosts.

Ticks are vectors of multiple disease-causing zoonotic pathogens and represent a significant threat to animal and human health [8]. It has been shown that tick burdens on animal hosts are highly aggregated, with a minority of hosts carrying most of the ticks [9,10]. This pattern implies that a small number of animal hosts are maintaining tick populations and thus contribute to natural transmission cycles of tick-borne pathogens, given that these hosts are most likely both to become infected and to be the source of infection for many uninfected ticks. Multiple factors have been postulated to be responsible for variations in tick burdens on their hosts such as host-intrinsic factors that are linked to individual physical (e.g. sex, age, body mass, body condition) [10–12], physiological (e.g. testosterone level) [13] and behavioural (e.g. foraging style) [14] traits, and host-extrinsic factors that are associated with environmental (e.g. vegetation cover and type) and climatic (e.g. humidity, temperature) attributes affecting host-seeking tick density and host density [11,15,16], together with abundance and activity of mammalian predators of hosts [17]. It is likely that tick burden patterns are driven by complex interactions with these host-intrinsic and -extrinsic factors [11,16]. Wildlife hosts are an essential component of the enzootic cycle of tick-borne pathogens [18] and thus any changes in host populations associated with landscape heterogeneity (i.e. landscape composition and configuration), intra- and interspecific interactions (e.g. competition, predation, parasitism) and climate [19], are likely to affect spatial and temporal dynamics of tick-borne diseases. Recently, using a mechanistic movement model, we identified the importance of tick dispersal by hosts in tick-borne pathogen spread dynamics in broad terms [20]. Consequently, there is a clear need to examine in more detail how host dispersal behaviour affects tick distribution patterns.

It has been recognized that the ecological process of dispersal (i.e. individual movement from a natal habitat patch to a new patch) is essential to understand the expanding distributions of species in response to anthropogenic environmental changes [21–23], including ticks and tick-borne pathogens [18]. In many spatial models projecting future trends in tick-borne diseases [24–26], host dispersal behaviour is linked to landscape characteristics in simplistic ways (e.g. modelled by a simple parameter) [27]. However, significant advances have been made in understanding and modelling dispersal mechanisms [28–30]. In particular, modelling the three context-dependent phases of the dispersal process, such as emigration, transfer (or interpatch movement) and settlement (or immigration) [27], is important for making reliable projections of spatio-temporal distributions of animal species [30–32]. In addition, inter-individual variability in host dispersal behaviour is often ignored in tick-borne disease models [24–26,33], most of which treat dispersal as species- or population-level average behaviour [34]. However, empirical studies showed that personality-related host behaviour (i.e. consistent individual differences in behaviour over time or from contexts [35]) can induce changes in a large range of ecological traits (e.g. habitat use, dispersal, fitness) [36–38], which can have consequences for tick parasitism [39,40]. For example, exploratory and bold hosts having larger home ranges and moving further from their natal site are more likely to encounter ticks, have high tick burdens and act as super-spreaders of ticks and tick-borne pathogens [39]. These individuals may have a central role in tick-borne pathogen transmission and spread processes [41], and should be considered in theoretical or applied epidemiological models of tick-borne diseases. To properly explore host dispersal, we need to develop spatially explicit and individual-based movement mechanistic models under the general conceptual framework of Nathan et al. [42]. The use of such models that integrate greater realism in

the dispersal process should lead to a better understanding of tick-host-pathogen interactions. Finally, movement mechanistic models should provide a useful tool to prioritize tick and tick-borne pathogen control actions by offering the possibility for managers to test different host- and/or environment-targeted control strategies and to identify what strategies would be effective in reducing tick-borne disease risk.

In this study, we built a stochastic and spatially explicit agent-based model (ABM) that includes both a mechanistic representation of host dispersal behaviour and inter-individual variability in dispersal for a tick-host-pathogen system, to explore how host dispersal behaviour affects infected tick burden patterns, while accounting for landscape heterogeneity (including fragmentation), the tick and host demographic processes and the host-specific pathogen transmission process. We conducted exploratory research by performing a global sensitivity analysis to assess the relative influence of ABM input parameters on infected tick burdens on hosts and infected tick distributions among hosts. These two ABM output variables determine local density of host-seeking infected ticks, which is used as an indicator of tick-borne disease risk [43]. We simulated fine-scale invasion of ticks by hypothetical host species varying in individual density, dispersal ability and lifespan across theoretical landscapes that differed in clumping and availability of habitat suitable for hosts and ticks.

# 2. Material and methods

## 2.1. Model overview

The ABM integrates three layers representing simulated host and tick populations, together with the landscape (figure 1). The interactions between the layers depended on four ecological processes: (i) tick population dynamics, (ii) host dispersal, (iii) host population dynamics and (iv) pathogen transmission. The ABM evolves according to discrete time steps of one day in a cell-based setting [44] in which the ecological processes associated with host dispersal behaviour and host population dynamics act at the cell scale. Consequently, the context dependencies operating in the dispersal process happen at this scale. Host individuals that are located in a given cell define a distinct open population (i.e. experiencing recruitment through births and immigrations, or losses through deaths and emigrations). Similarly to Li *et al.* [33], we considered a set of gridded theoretical landscapes of 50 (rows) × 50 (columns) cells with a cell resolution of 100 m and varying in clumping and proportion of breeding habitat (electronic supplementary material, Appendix A). The ABM was parameterized from empirical and modelling studies, together with expert opinion on the bacterium *Borrelia burgdorferi sensu stricto*, the black-legged tick (*Ixodes scapularis*) and hypothetical reservoir hosts (electronic supplementary material, Appendix B). The latter represent a broad range of avian and terrestrial animal species with realistic characteristics, including dispersal ability, individual density and lifespan. The ABM can be adapted for any species or regions with sufficient empirical data as a wide set of input parameters are integrated in the model. A detailed description of our ABM following the Overview, Design concepts and Details) protocol [45–47] is provided in the electronic supplementary material, Appendix A.

### 2.1.1. Tick population dynamics

The life cycle of ixodid (or hard-bodied) ticks consists of four continuous development stages: eggs, larvae, nymphs and adults [48]. In each post-egg stage, ticks are either in the questing phase (i.e. climbing up vegetation and waiting for passing hosts, or waiting until microclimatic conditions are suitable for activity), attached to hosts or successfully fed to repletion and detached from hosts [49]. In general, questing ticks attach to hosts for blood meals after encountering host individuals [50]. Once feeding is complete, engorged ticks detach from their hosts and fall to the ground where they overwinter and moult into the next life stage [49]. Adult female ticks require blood meals to lay eggs that will hatch into larvae [51]. In addition, migratory birds can transport ticks and pathogens over long distances to new non-endemic areas, and can thus contribute to range expansion of both ticks and tick-borne pathogens [52–54]. For example, migratory passerine birds have facilitated northward range expansion of *I. scapularis* ticks and the bacterial cause of Lyme disease, *B. burgdorferi*, in North America [55–57] during their spring migration, by virtue of their role as tick hosts and reservoir hosts for tick-borne pathogens. Consequently, we considered four key processes to model tick population dynamics: tick questing activity, tick attachment to hosts, tick detachment from hosts and tick

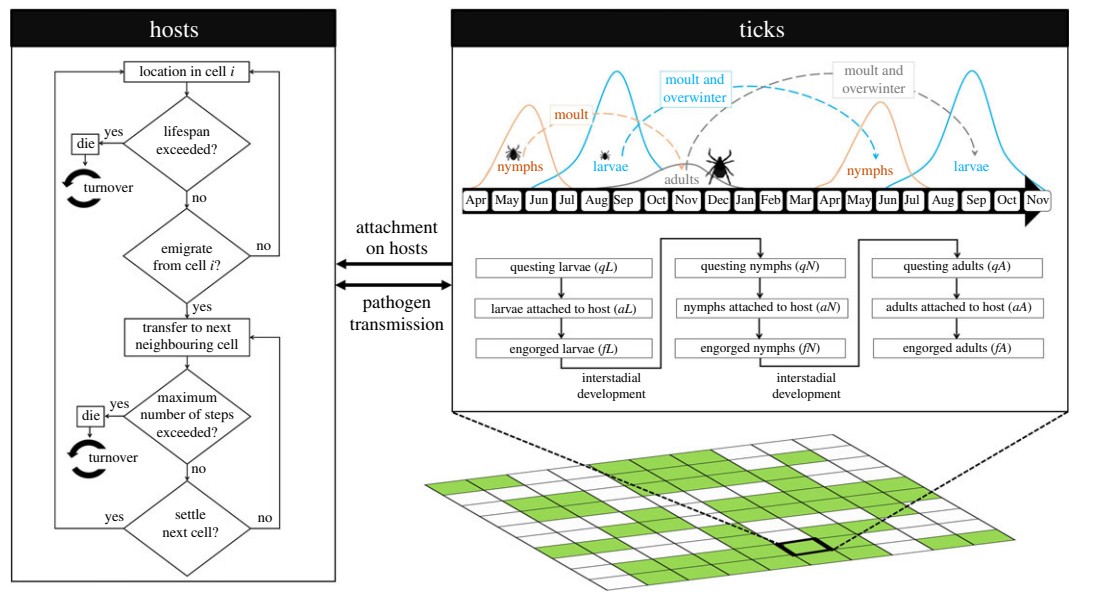

**Figure 1.** Conceptual framework of the spatially explicit ABM showing the three layers (i.e. simulated host and tick populations, together with the landscape). For the host layer, the rectangles represent actions and diamonds define decision points. Within a single time step (one day), simulated hosts can move to a cell according to dispersal rules, then the tick population distribution is updated and finally the possibility of pathogen transmission is evaluated.

introduction by migratory birds. The phenology of ticks was programmed to be influenced by seasons, host availability and habitat types (electronic supplementary material, Appendix A).

### 2.1.2. Host dispersal

Using the general framework described by Travis *et al.* [30] for modelling individual dispersal, the host dispersal process in our ABM involves three phases: emigration, transfer and settlement. In the emigration phase, each host has a probability of leaving its natal cell at a given discrete time step and this probability varied as a function of the presence of conspecifics in the cell. The settlement decision, which is defined by the probability that a host stops in a new cell, depended upon both the habitat type and the number of conspecifics present in the cell at a given time step. We modelled the transfer phase (i.e. host movement among landscape cells) with a stochastic movement simulator [58] by incorporating inter-individual variability in dispersal behaviour [34]. At each time step, all dispersing hosts move simultaneously to one of the eight adjacent cells from their current cell (i.e. they make one move). The choice of the next cell depended on three parameters: (i) movement costs through the landscape cells within the individual's perceptual range, (ii) distance to the current cell (i.e. equal to $\sqrt{2}$ for diagonal cells and 1 otherwise), and (iii) individual's directional persistence, which corresponds to the tendency for successive movement steps to be in the same direction. A selection probability for each of eight neighbouring cells was calculated as the reciprocal of their 'effective cost' of moving, which was given by the harmonic mean of the movement costs of all cells within the individual's perceptual range. The host moves to the cell with the highest selection probability. Finally, we applied a cost adjustment parameter to the cost of each cell to obtain an adjusted cost, which allowed simulating individuals with different personality types (i.e. 'shy' versus 'bold'). More details can be found in the electronic supplementary material, Appendix A.

### 2.1.3. Host population dynamics

We did not consider demographic processes related to reproduction in our ABM, but instead, we modelled a turnover process of host populations [59], which allows maintaining demographic stability of populations and preventing all individuals from becoming infected at the end of each simulation. When a simulated host exceeded its lifespan or died owing to dispersal mortality, the host was replaced by another conspecific that was cleared of infection and randomly placed in a habitat cell occupied by host individuals. We assumed that birth and death rates are equal and vary over time. The demographic turnover process in the ABM is a simplified representation of the individual recruitment process observed in reality. In some locations where oak trees are abundant, rodent

populations can experience boom-bust cycles of growth in response to episodic production of acorns [60]. In this study, we did not simulate seasonally or inter-annually fluctuating resources, but instead, we ran simulations across spatially heterogeneous landscapes varying in clumping and availability of suitable habitat. The effects of resource pulses could be explored in future studies. In the absence of such effects, it was assumed that host populations are demographically stable given that wildlife host fitness (i.e. survival and reproduction) is not significantly affected, to our current knowledge, by high tick burdens [61] and the bacterium *B. burgdorferi* [62].

### 2.1.4. Pathogen transmission

Generally, three possible routes of pathogen transmission between ticks and reservoir hosts are recognized in tick-borne diseases [63]: (i) systemic, occurring when the pathogen is acquired by an uninfected tick from an infected host or by an uninfected host from an infected tick during a blood meal, (ii) non-systemic, taking place when the pathogen is transmitted between co-feeding uninfected and infected ticks that feed simultaneously on the same host in close proximity, and (iii) transovarial, occurring when an adult female tick acquires the infection during a blood meal and transmits the pathogen to its eggs. Because transovarial transmission is rare or non-existent for the bacterium *B. burgdorferi* [64] and the role of non-systemic transmission in the maintenance of the bacterium in nature remains unclear [65], we only considered systemic transmission (see the electronic supplementary material, Appendix A for more details). We assumed that hosts cannot clear their infection and remain infected for life [66].

## 2.2. Model implementation

The ABM was implemented in the R statistical software (v. 3.6.1) [67] using the NetLogoR package [68]. All simulations were performed over a total period of 10 years given that our assumptions of geographically closed host populations from outside the study area (i.e. not experiencing recruitment or losses) help to maintain stable populations. Uninfected hosts were randomly distributed in the breeding habitat cells of the landscape with a number of individuals equal to $K^H$. Hosts were active from February to November. At each year, uninfected and infected ticks were transported by migratory passerine birds into the landscape cells during the spring migration period (1 May–30 June). We did not explicitly model northward spring migration, but instead, migration provided a supply of ticks to approximate the effects of tick introduction by migratory birds. After the tick introduction by migratory birds, questing tick population dynamics was updated within the landscape cells at each time step. Tick attachment to hosts was then considered and pathogen transmission was evaluated between reservoir hosts and ticks. Finally, attached ticks fell off hosts at the end of the time step.

## 2.3. Sensitivity analysis

A global sensitivity analysis (SA) was performed to assess the sensitivity of simulated infected tick burden patterns to variations in ABM input parameters. The SA allowed the identification of host-intrinsic and -extrinsic parameters that may be those among the most influential on tick-borne disease risk, and those for which research effort should be prioritized to better understand the ecological processes underlying tick-borne disease risk. We included 56 input parameters in the SA. A Latin hypercube sampling scheme was used to sample 200 different parameter combinations from a uniform probability distribution, satisfying the minimum value of $4N/3$, where $N$ is the number of input parameters [69]. Because our ABM included stochasticity in the ecological processes and the construction of the landscapes, 10 replicated landscapes were generated for each parameter combination, resulting in a total of 2000 simulations.

Similarly to our previous paper [20], we built boosted regression tree (BRT) models [70] to assess the relative influence of each input parameter on ABM output variables and to identify which of the parameters had the highest effect on these output variables. The average burden of infected ticks per host (or average number of infected ticks per host individual) for each post-egg live stage (i.e. larvae, nymphs and adults) and the Hoover concentration index (0–1) were defined as output variables. We used the Hoover concentration index to quantify heterogeneity in infected tick burdens among hosts [71]. This index takes the value of zero when all hosts have the same infected tick burden and the value of one when all infected ticks are concentrated on a single host. High values for the Hoover concentration index thus reveal the presence of super-spreaders of infected ticks. The output variables were measured within an area that excluded the 10 rows of cells on the sides of the study area to minimize edge effects. We chose a Gaussian error structure for the loss function. The average burdens

of infected ticks per host were log-transformed in the BRT models to stabilize the variance and to meet the assumption of Gaussian error distributions [72]. Several combinations of learning rate (0.01, 0.005, 0.001), tree complexity (1–5) and bag fraction (0.5, 0.7, 0.9) were tested to define optimal settings for the BRT models. The parameter combination with the lowest 10-fold cross-validation deviance was selected to fit the final BRT model. We also evaluated the relative influence of each ecological process that is integrated into the ABM. We used partial dependence plots in which 95% confidence intervals were obtained from 500 bootstrap replicates to visualize the relationships between the output variables and the most influential input parameters. The significance of the strongest interactions was tested using 100 bootstrap resampling iterations [73,74]. The BRT models were built using the *dismo* and *ggBRT* packages in R [74,75].

# 3. Results

Our BRT models showed high performance in quantifying the relative influence of each ABM input parameter on the infected tick burden on hosts and the infected tick distribution among hosts with explained 10-fold cross-validation deviance ranging from 92% to 99% and with 10-fold cross-validation correlation ranging from 0.96 to 0.99 (electronic supplementary material, table C1 in Appendix C). In the sections below, we report the relationships between the ABM output variables and the most influential parameters ($\geq 10\%$ relative influence). The relative influence values of all input parameters are in the electronic supplementary material, Appendix C (figures C1 and C2).

## 3.1. Infected tick burden on hosts

The average burden of infected larvae per host was mainly influenced by input parameters related to the pathogen transmission process (40% relative influence), together with the tick attachment and detachment processes (32% relative influence; electronic supplementary material, table C2 in Appendix C). The probability of pathogen transmission from an infected host to an uninfected larva was the main predictor of the output variable, with a positive nonlinear relationship for the predictor that displayed a threshold level around 0.75 ($\beta^{HL}$; 39% relative influence; figure 2a). This relationship is expected as host-to-tick transmission efficiency is a key determinant of tick-borne disease risk. There was also a negative nonlinear relationship for the slope at the inflection point of the on-host density-dependent larva mortality function ($\alpha_d^{aL}$; 28% relative influence; figure 2a). This predictor controls the density-dependent mortality rate of larvae attached to hosts. The decrease in the average burden of infected larvae per host was rapid when the slope at the inflection point of the on-host density-dependent larva mortality function was low ($\leq 0.02$), and became increasingly slow as the slope increased. The strongest pairwise interaction was high (interaction size: 31.41) and significant ($p < 0.05$), reflecting the interacting effects of the most influential predictors, with infected larva burdens being highest when the probability of pathogen transmission from an infected host to an uninfected larva was high and the density-dependent mortality rate of larvae attached to hosts was low (electronic supplementary material, figure C3 in Appendix C). The other interactions between predictors were weak (interaction size < 3).

The average burden of infected nymphs per host was primarily influenced by input parameters related to the tick questing activity process (36% relative influence), together with the tick attachment and detachment processes (34% relative influence; electronic supplementary material, table C2 in Appendix C). The average burden of infected nymphs per host showed a positive nonlinear relationship with increasing the base host-finding rate of questing nymphs ($a_0^{NH}$; 22% relative influence; figure 2b) that displayed a threshold level around 0.006 d$^{-1}$. In addition, the output variable was negatively correlated with the mortality rate of ticks developing from engorged larvae into questing nymphs in breeding habitats ($d^{fL}$; 20% relative influence; figure 2b), indicating higher infected nymph burdens on hosts when the developing tick mortality rates were low ($\leq 0.00125\,d^{-1}$). The strongest pairwise interaction was very weak (interaction size = 0.98) and not significant ($p > 0.05$), reflecting the additive effects of the most influential predictors.

The average burden of infected adult ticks per host was mainly influenced by input parameters related to the tick attachment and detachment processes (29% relative influence), together with the tick questing activity process (25% relative influence; electronic supplementary material, table C2 in Appendix C). Three key predictors contributed to the average burden of infected adult ticks per host: the base host-finding rate of questing adult ticks ($a_0^{AH}$; 22% relative influence; figure 2c), the proportion of migratory passerine birds carrying nymphs ($p_N^B$; 13% relative influence; figure 2c) and the shape parameter of the

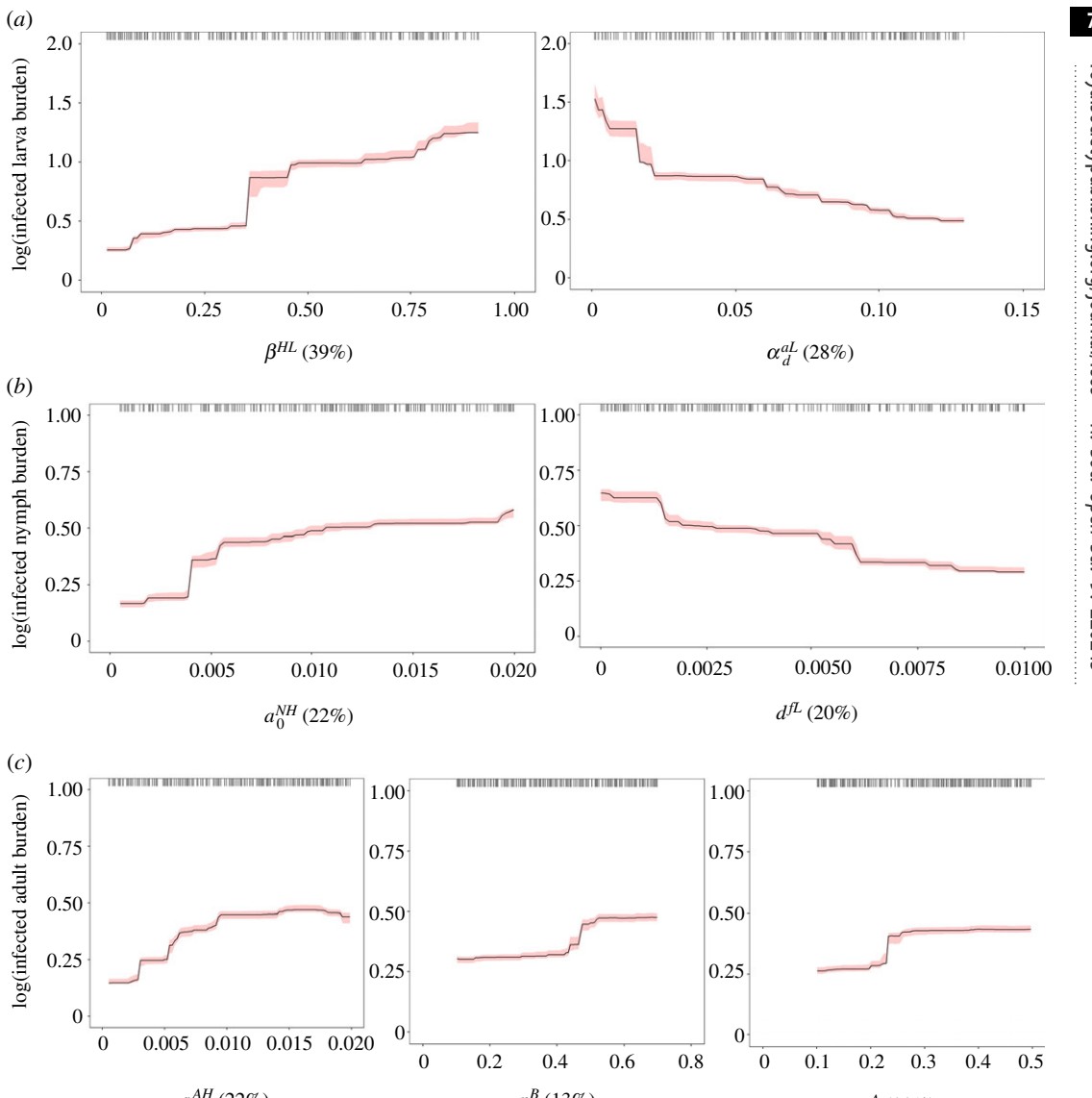

**Figure 2.** Partial dependency plots with bootstrapped 95% confidence intervals (red) for the most influential input parameters ($\geq$ 10% relative influence) predicting the average burden of infected larval ticks per host (*a*), the average burden of infected nymphal ticks per host (*b*), and the average burden of infected adult ticks per host (*c*). The average burdens of infected ticks per host were log-transformed. Black tick marks at the top of each plot represent raw data. Relative influence (%) of each input parameter is indicated in parentheses. $\beta^{HL}$: probability of pathogen transmission from an infected host to an uninfected larva; $\alpha_d^{aL}$: parameter that controls the density-dependent mortality rate of larvae attached to hosts; $a_0^{NH}$: base host-finding rate of questing nymphs; $d^{fL}$: mortality rate of ticks developing from engorged larvae into questing nymphs in breeding habitats; $a_0^{AH}$: base host-finding rate of questing adult ticks; $p_N^B$: proportion of migratory passerine birds carrying nymphs; $\sigma^A$: parameter that controls the width of the adult tick questing activity peak.

phenomenological curve of adult tick questing activity, which controls the activity peak width ($\sigma^A$; 11% relative influence; figure 2*c*). The three predictors displayed a positive nonlinear relationship with the output variable. Threshold levels for high burdens of infected adult ticks occurred around 0.009 d$^{-1}$, 0.5 and 0.25, respectively. The strongest pairwise interaction was very weak (interaction size = 0.73) and not significant ($p > 0.05$), reflecting the additive effects of the most influential predictors.

## 3.2. Infected tick distribution among hosts

The infected tick distribution among hosts was mostly influenced by input parameters related to the host dispersal process (44% relative influence for larvae, 43% relative influence for nymphs and 47% relative

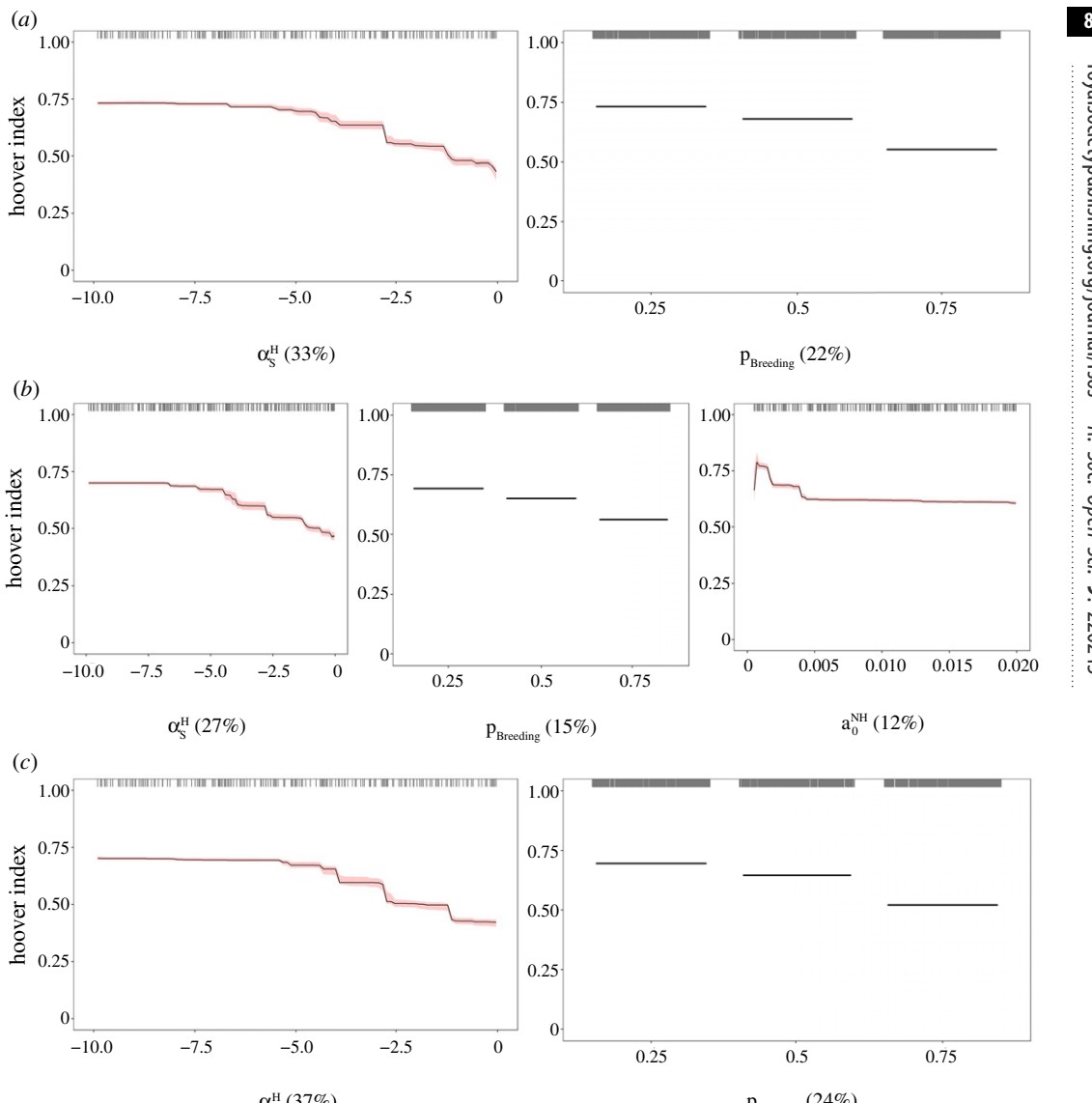

**Figure 3.** Partial dependency plots with bootstrapped 95% confidence intervals (red) for the most influential input parameters (≥ 10% relative influence) predicting the infected larval tick distribution among hosts (*a*), the infected nymphal tick distribution among hosts (*b*), and the infected adult tick distribution among hosts (*c*). Black tick marks at the top of each plot represent raw data. Relative influence (%) of each input parameter is indicated in parentheses. $\alpha_s^H$: parameter that controls the effects of conspecific density on host settlement decisions; $p_{Breeding}$: proportion of breeding habitat; $a_0^{NH}$: base host-finding rate of questing nymphs.

influence for adult ticks; electronic supplementary material, table C2 in Appendix C). The infected tick distribution among hosts was best described by a negative nonlinear relationship with the slope at the inflection point of the density-dependent settlement function, which controls the effects of conspecific density on host settlement decisions ($\alpha_s^H$; 33% relative influence for larvae, 27% relative influence for nymphs and 37% relative influence for adult ticks). This suggests that a settlement strategy with negative density dependence would promote super-spreading of infected ticks (figure 3). In our simulations, a threshold level around −4 was required to induce high heterogeneity in infected tick burdens among hosts. The proportion of breeding habitat was the second strongest predictor ($p_{Breeding}$; 22% relative influence for larvae, 15% relative influence for nymphs and 24% relative influence for adult ticks) and was negatively correlated with the Hoover index (figure 3), suggesting that hosts are more likely to act as super-spreaders of infected ticks in highly fragmented landscapes with a small proportion of breeding habitat (≤0.25). Finally, a negative nonlinear relationship was also observed between the Hoover index and the base host-finding rate of questing nymphs ($a_0^{NH}$; 12% relative

influence; figure 3*b*). This suggests that super-spreaders of infected nymphs are more likely to be found in areas where encounter rates with questing nymphs are low. A threshold level for high heterogeneity in infected nymph burdens among hosts occurred around $0.004\,\mathrm{d}^{-1}$. For each final BRT model, the strongest pairwise interaction was very weak (interaction size < 0.4) and not significant ($p > 0.05$), reflecting the additive effects of the most influential predictors.

# 4. Discussion

In the face of increasing public health and economic impacts of tick-borne diseases worldwide [76], there is growing awareness of the importance of identifying ecological drivers that contribute to heterogeneity in tick burdens among hosts to improve our capacity to predict and control tick-borne disease risk [43]. While the increased risk has been attributed to a range of host-intrinsic and -extrinsic factors (e.g. [11,16]), the relative importance of context-dependent host dispersal behaviour is often unknown. In a previous study, we built a 'reaction-advection-diffusion' type model to simulate northward invasion of ticks by migratory birds and terrestrial hosts over large spatial scales across spatially heterogeneous landscapes varying in resource aggregation [20]. This population-level model with a continuous representation of space and time integrated a mechanistic formulation of host movement to quantify the relative importance of ecological drivers in predicting global spread of infected ticks, tick infection prevalence and infected tick density, and identified host dispersal as an important determinant of tick-borne disease spread. Here, we explored in detail key aspects of host dispersal ecology, in accordance with emerging knowledge of this field of ecology. This was achieved using a spatially explicit ABM with a discretized representation of space and time, which allowed modelling fine-scale individual-level movement. The ABM explicitly considered the dispersal process in its three phases of emigration, transfer and settlement, and accounted for inter-individual variability in dispersal. Our simulations indicated that ecological predictors of infected tick burdens on hosts differed among the post-egg life stages of ticks (i.e. larvae, nymphs and adults), with tick attachment and detachment, tick questing activity and pathogen transmission dynamics identified as key processes, in a way consistent with our knowledge of tick-borne disease ecology. These results demonstrate that the ABM has the potential to reproduce patterns observed in real systems. We also found that the type of host settlement strategy and the proportion of habitat suitable for hosts determined heterogeneity in infected tick burdens among hosts. These findings highlight the importance of context-dependent host dispersal and habitat fragmentation in super-spreading of infected ticks.

## 4.1. Infected tick burden on hosts

Our results highlight the dual influences of the pathogen transmission and tick attachment-detachment processes on the infected larval tick burden on hosts. We found that high probabilities of pathogen transmission from an infected host to an uninfected larva ($\geq 0.75$) and low density-dependent mortality rates of larvae attached to hosts ($\leq 0.007\,\mathrm{d}^{-1}$) resulted in high burdens of infected larvae on hosts. Therefore, variations in infected larva burdens would be attributed to changes in both reservoir competence of hosts for the pathogen and host grooming behaviour, as seen in nature [77]. As expected, the density-dependent mortality rate of larvae feeding on hosts decreased with increasing the infected larva burden. An increase in tick burden can induce a reduction in feeding success owing to an increase in host grooming behaviour [78] or acquired immunological resistance of hosts [79], while a decrease in tick burden would reduce grooming behaviour or acquired immunological resistance and thus would favour survival of ticks feeding on hosts [43]. Several host species differ in their ability to successfully feed many ticks and to infect feeding ticks with a pathogen [80]. Mice are considered as the most competent host species for the bacterium *B. burgdorferi* and are highly permissive to larva feeding [80], with almost 50% of larvae fed to repletion [81]. By contrast, 83–96% of larval ticks can be killed by alternative host species (e.g. squirrels), which can be characterized by high larval burdens and low reservoir competence [81].

Tick questing activity and tick attachment-detachment were among the most influential processes predicting the infected nymph burden on hosts. Logically, from the results, hosts carried higher body burdens of infected nymphs in areas with high encounter rates with questing nymphs ($\geq 0.006\,\mathrm{d}^{-1}$) and low mortality rates of ticks developing from engorged larvae into questing nymphs in breeding habitats ($\leq 0.00125\,\mathrm{d}^{-1}$). The latter parameter controls the questing nymph populations in the

landscape by affecting the proportion of engorged larvae that moult into questing nymphs during the interstadial development phase, and field studies show that high tick burdens on small-mammal hosts coincide with high densities of host-seeking ticks on vegetation [11,82]. Tick mortality during the interstadial development phase is largely influenced by temperature and humidity (reviewed in [83]), with extremes of temperature, drowning and desiccation being the main determinants of tick mortality. Climate also determines tick questing activity (reviewed in [84]) and thus impacts on encounter rates between hosts and ticks, which can experience different mortality rates according to temperature conditions. We did not integrate input parameters related to climate in the ABM, although temperature and humidity would affect tick burdens on hosts because of their direct effects on tick activity and survival [84].

Like nymphal ticks, the infected adult tick burden on hosts was logically related to the tick attachment-detachment and tick questing activity processes. Higher infected adult tick burdens on hosts occurred in areas where encounter rates with questing adult ticks were high ($\geq 0.009 \, \mathrm{d}^{-1}$), the proportion of migratory passerine birds depositing engorged nymphs in the landscape habitat cells was large ($\geq 0.5$) and the duration of adult tick questing activity was long ($\geq 30 \, \mathrm{days}$). The prolonged questing activity would increase feeding opportunities for adult ticks to find a host. The results suggest, logically, that changes in both the duration of adult tick questing activity and encounter rates with hosts would have important impacts on infected adult tick burdens. Our results also highlight the importance of considering migratory birds in explaining infected adult tick burdens on hosts. Birds can contribute to *B. burgdorferi* transmission by dispersing infected larval and nymphal ticks into areas where infected ticks can transmit the bacterium to uninfected hosts or by favouring the establishment of new populations of ticks into suitable habitats [56,85]. Ogden *et al.* [57] identified that approximately 2% of *I. scapularis* tick-infested migratory passerine birds that were located at stopovers in southeastern Canada, carried mostly nymphal ticks and estimated that 50 million to 175 million immature *I. scapularis* ticks could be transported by migratory passerine birds across Canada during each spring.

## 4.2. Infected tick distribution among hosts

Our results highlight the importance of host dispersal behaviour in infected tick distribution patterns among hosts. We found that the type of host settlement strategy and landscape characteristics, specifically the proportion of habitat suitable for hosts, strongly determined heterogeneity in infected tick burdens among hosts. Low encounter rates with questing nymphs ($\leq 0.004 \, \mathrm{d}^{-1}$) also promoted heterogeneity in infected nymph burden patterns, but to a lesser extent. This heterogeneity occurred in areas with low host individual densities, in which a few hosts fed the majority of ticks. From our results, super-spreaders of infected ticks, i.e. host individuals that have disproportionate burdens of infected ticks, were more likely to be present in highly fragmented landscapes with low levels of habitat suitable for hosts ($\leq 0.25$) and to have settlement strategies that rely on both finding suitable habitat while avoiding cells with high densities of conspecifics. The empirical evidence or modelling assumptions supporting our results are, to date, absent. On the basis of other studies, we consider that the explanation of observed patterns is related to the effects of context-dependent settlement behaviour and landscape characteristics on the rate of host spread [28,86]. For example, the modelling study of Bocedi *et al.* [28] revealed that settlement strategies involving either only finding suitable habitat or finding suitable habitat with negative density dependence induced higher population spread rates compared to density-independent settlement strategies, with higher differences observed in landscapes with low rather than high amount of suitable habitat. These results suggest the existence of a 'shadow effect', a phenomenon that happens when suitable habitat patches intercept individuals dispersing through the matrix, compelling them to stop in cells close to their natal cell, and thus decrease the probability of dispersing towards other patches, with the consequence of decreasing the spread rates [87]. Ultimately, settlement strategies with negative density dependence should be incorporated in spatial models of tick-borne diseases to predict the infection risk of these diseases.

## 4.3. Implication for controlling tick-borne diseases

Identifying super-spreaders is of key importance for reducing infectious disease risk [4,88]. The efficiency of tick-borne disease control could thus be improved by identifying super-spreaders of infected ticks. Our findings suggest that host individuals using settlement strategies with negative density dependence (i.e. individuals that are less likely to settle a habitat cell containing high density of conspecifics) and thus exhibiting active dispersal would have more chance of acting as super-spreaders of infected ticks. We

also found that highly fragmented landscapes with a small proportion of habitat suitable for hosts ($\leq 0.25$) would be more likely to promote super-spreading of infected ticks. These findings are of importance for tick-borne disease control because super-spreading of infected engorged larvae has a strong impact on tick-borne pathogen transmission. However, further work that extents the ABM is required for testing the efficiency of control methods targeting super-spreaders of infected ticks.

The ABM presented in this study describes the transmission cycle of a single infectious pathogen circulating within tick populations and a single host species. However, ticks are often generalist vectors feeding on a large variety of vertebrate hosts with different body sizes, including mammals, birds and reptiles [89]. In particular, interspecific interactions (e.g. competition and predation) can affect tick-borne disease dynamics [17,90]. For example, it has been demonstrated that predation can regulate tick populations either by reducing density of amplification hosts (i.e. hosts that can infect many ticks such as rodents) in the environment [91,92] or via antipredator behaviour of amplification hosts, which reduce their activity and movement in response to direct cues of predator presence [93]. Our ABM could be extended to more ecologically complex systems with the incorporation of multiple species and real landscape complexity to test different host- and/or environment-targeted intervention strategies and to identify what strategies would be effective in reducing tick-borne pathogen infection risk for humans. An application of this extended ABM to tick control strategies could thus greatly improve planning on-the-ground public health interventions.

Data accessibility. The data and code are available from: http://doi.org/10.5061/dryad.nzs7h44rx [94].
Authors' contributions. O.T.: conceptualization, data curation, formal analysis, investigation, methodology, project administration, resources, software, supervision, validation, visualization, writing—original draft, writing—review and editing; C.E.V.: supervision, validation, writing—review and editing; C.B.: conceptualization, funding acquisition, project administration, supervision, validation, writing—review and editing; N.H.O.: conceptualization, funding acquisition, project administration, supervision, validation, writing—review and editing; P.A.L.: conceptualization, funding acquisition, project administration, supervision, validation, writing—review and editing.

All authors gave final approval for publication and agreed to be held accountable for the work performed therein.
Competing interests. We declare we have no competing interests.
Funding. This work was funded by the Public Health Agency of Canada.
Acknowledgements. We are grateful for the computational resources provided by Calcul Québec (https://www.calculquebec.ca/) and Compute Canada (https://www.computecanada.ca/). All simulations were run on the Cedar supercomputer (Simon Fraser University), which is managed by Compute Canada. We would like to thank Daniel Stubbs for the excellent technical support. We also thank Hsiao-Hsuan Wang and one anonymous reviewer for their constructive and useful comments, which helped to greatly improve the manuscript.

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
