## [Peer Review File · Royal Society Open Science]

Review History

RSOS-211380.R0 (Original submission)

Review form: Reviewer 1

Is the manuscript scientifically sound in its present form?

Yes

Are the interpretations and conclusions justified by the results?

No

Is the language acceptable?

Yes

Do you have any ethical concerns with this paper?

No

Have you any concerns about statistical analyses in this paper?

No

Recommendation?

Major revision is needed (please make suggestions in comments)

Comments to the Author(s)

This is an interesting paper exploring the role of host dispersal, landscape characteristics, and pathogen infection process impact tick-borne pathogens in modelled landscapes. Appendices are very clear and well documented. Data and code and supporting documentation are clear and well organized.

However, the authors recently published a paper with a similar title ("Mechanistic movement models reveal ecological drivers of tick-borne pathogen spread" versus this one "Ecological drivers of tick-borne pathogen infection risk: Inference from a mechanistic modelling approach of host dispersal"). They also discuss similar themes in the abstract and intro in both motivations, analytical approaches and some of the results in terms of figures made. Yet it is thus surprising they do not engage with their own work in the discussion. It's critical that the authors make clear - in all aspects of the paper - what the unique contributions of this follow up paper are and how the findings are both different and also contributing information that the previous paper did not. Additionally, the introduction sets up particular ideas about what will be included in the model and why - but the documentation as is in the methods and then discussion could make clearer for readers what mechanisms are potentially at play and how different assumptions could influence the findings. The authors make bold claims about what their results demonstrate - but it is still a model. More discussion of how the findings fit within empirical field research would be useful, as well as introducing more of the specifics of the system in the introduction and methods, as they greatly shape the model and study design, rather than the end of the discussion.

Line comments

Abstract

20 - "worrying rate" is not specific - what does this mean and to whom? Clearer alternatives could be as simple as "sharply increasing"

21 - be clear from the get go whether public health means people only, 'one health' model of people, ecosystems, and animal health, etc.

25 - does 'host' here mean non-human animal hosts? Or including people?

Introduction

54 - "should" - would be clearer to say that "this pattern implies a small number of hosts are maintaining tick populations and thus contribute..."

56 - be clearer - are you discussing only animal hosts here? Or people too? Because becomes important when you then list the individual characteristics

66 - 67 - suggest re-organizing the paragraph or starting with a different topic sentence. Jumps from dispersal to landscape connectivity. Need to make this paragraph flow of ideas connect more to each other for the reader. The next paragraph discussing habitat clumping, availability and quality, but the importance of those - specifically those elements of landscapes and for tick systems - is not introduced in the introduction before this

67 "animal species invasion" - unclear. Are you referring to predicting invasive animal species' establishment or movement into new areas? Or just discussing movement/establishment generally and not specifically invasions?

72 - 74 - says studies don't do X, but does not cite anything to demonstrate that (from a paper discussing needed research, for example) or providing examples of what such studies do include

74 - 75 = there are other differences between individuals that may explain their movement in the landscape that are not just about "boldness" or fear in movement ecology - why focus on that here?

78 - 83 = no hypotheses? The introduction seems to indicate that you believe you will find certain patterns, based on mechanisms discussed

Methods

89 - 90 = "host dispersal" - is this being used interchangeably with behavior? Questing distance as measure of boldness? Why? Are there other studies or systems that do this?

107 - 113 = sources for tick natural history information?

141 - 147 = great to be clear about what and was not considered. Can you please add a bit more to explain how this realistically reflects life history / life cycles and provide citations?

162 - please cite software and then cite the individual packages; they have distinct authors and citations

162 - 164 - please explain more about this closed host population and temporal assumptions - what kind of organism(s) are you imagining that this would be appropriate for?

166 - 168 = spring migration where? Transported where?

190 - why transformed?

Results

Figure 3 - consider larger fonts for axes

Discussion

296 - 298 - what makes this model difference from the model in your previous, related paper?

299 - 303 - how do these findings contribute new information in ways that your recent, related paper did not?

306 - 307 = how do you think that your model parameters about tick detachment shape your conclusions? How may they change if you changed those parameters?

311 - how does host grooming show up in your model?

315 - discussion of common hosts - especially if model was made with particular terrestrial hosts in mind - should be in the intro and methods, not introduced in the discussion

319 -

331 - here you discuss the role of climate in questing, not boldness/personality. This seems different than what the introduction was going to explore - in terms of understanding individual differences in movement. Can you say more in the discussion to explain how you believe all of these factors are at play? Can you separate them based on your design?

345 - 349 - discussion of the importance of migratory birds should come up in the intro and methods, since it shaped model and assumption development, not just the discussion

356 - 359 - so is your spatial model 'set' in Canada in terms of seasonal assumptions, etc? That is not discussed in the methods or intro, would be useful to understand this context

361 - this section is named 'heterogeneity in tick burden among hosts' but the 362 - 376 is discussing landscape dynamics, consider more explicit discussion of landscape characteristics as a section in both the results and discussion, especially since it's highlighted in the introduction as motivating the study question and design. Then critical to explore how this compares to existing empirical studies with different landscape contexts - and to reflect on how different parameters would shape the results found

362 - 369 - these results are modeled - how do they compare to empirical studies in this system?

Review form: Reviewer 2

Is the manuscript scientifically sound in its present form?

Yes

Are the interpretations and conclusions justified by the results?

Yes

Is the language acceptable?

Yes

Do you have any ethical concerns with this paper?

No

Have you any concerns about statistical analyses in this paper?

No

Recommendation?

Major revision is needed (please make suggestions in comments)

Comments to the Author(s)

This is an interesting, basically well-written paper, which merits publication. However, I am concerned that the authors have “oversold” their model regarding the context within which the current results provide useful information. I feel uncomfortable with the statements like one on lines 390-391: “Importantly, the application of such models (as ours) to tick control strategies could greatly improve planning public health interventions.” Admittedly, in the preceding lines (378-386) the authors point out important processes they have chosen to omit from the present model, all of which seem like reasonable simplifications. They then claim their model could easily be extended to include multiple species and real landscape complexity to test various hypotheses regarding tick-borne pathogen infection dynamics (lines 386-389). This also is fine, and I look forward to reading about the results of such model extensions and the associated sensitivity analyses. But the leap from results of such analyses to planning on-the-ground public health interventions is a big one. To quote the often-quoted Science article by Oreskes et al. (1994): “In areas where public policy and public safety are at stake, the burden is on the modeler to demonstrate the degree of correspondence between the model and the material world it seeks to represent and to delineate the limits of that correspondence.”

Another concern regarding the presentation/discussion of model results involves statements such as that on lines 307-310: “We found that high probabilities of pathogen transmission from an infectious host to an uninfected larva and low density-dependent mortality rates of larvae attached to hosts resulted in large infected larva burdens per host.” This result, per se, can be determined logically without looking at model results. Likewise, low probability of pathogen transmission from an infectious host to an uninfected larva and high density-dependent mortality rates of larvae attached to hosts logically results in relatively small infected larva burdens per host. Results from previous studies (lines 311-318) suggest that real-world host species differ regarding their likelihood of infecting ticks and regarding on-host tick survival rates. Thus, within this context, it seems that a legitimate question to ask is: “what new knowledge is embodied in these model results?” Or, to paraphrase lines 32-35 as a question: “How (specifically) does the model’s theoretical mechanistic framework provide a better understanding of the tick-borne pathogen infection risk and how (specifically) can it serve as the foundation for applied studies of scenario analyses on public health intervention strategies?” It seems to me that the authors are obliged to provide a clear and direct answer to that question. It also seems to me that assessing the relative influence of the various processes involved in tick-borne pathogen infection dynamics based on a hypothetical “composite” host species, although an interesting and appropriate model evaluation exercise, has limited utility in terms of scenario analyses of public health intervention strategies.

Of much more utility would be an exploration of threshold levels of host-to-tick pathogen transmission rates, on-host and off-host tick survival rates, and host encounter rates that would sustain dangerously high pathogen infection risks to humans. As I understand the model, it certainly has the capability to support such an exploration. For example, given the current parameterization of the model, how high do host encounter rates need to be to sustain dangerously high infection risk (measured as nymphal infection prevalence or some other appropriate metric)? And given that threshold level, what might be feasible habitat modifications or host population control interventions that could maintain host encounter rates

below that threshold? Perhaps I am being overly critical, but it is only because I see great unrealized potential regarding the use of the present model. If running and analyzing more simulations is beyond current possibilities, it would be useful to provide at least a description of what the type of model application I suggest might look like within the context of assessing possible public health intervention options.

I have looked at the model equations presented in the Supplementary Material but I have not looked at the model code. The representation of some key processes is phenomenological (e.g., tick questing activity as a function of day of year) and the representation of other key processes is theoretical (e.g., the representation of host dispersal using a non-linear density-dependent emigration function). How one defines “mechanistic modelling,” or, more generally, what one perceives as a cause-effect relationship, obviously is subjective. But I wonder if the use of descriptors like “mechanistic modeling approach” and “theoretical mechanistic framework” serves to clarify or confuse. It seems to me that “spatially-explicit and agent-based,” without sprinkling in the additional descriptors, conveys more clearly the type of model that is presented in the paper.

Just a couple of other items caught my eye. The term “supershedding events” is mentioned in the Abstract and a couple of times in the Discussion, but not in the Results. It would be useful to link that term directly to the results. Implicitly, hosts with high tick burdens potentially re supershedders. But which results indicate the potential presence of supershedders? The only presentation of tick burdens in terms of ticks per host that I can find is in Figure 2, but I cannot relate those partial dependency plots to likelihood of supershedding events without additional guidance. Finally, the journal name is missing on line 521 and on line 527.

My best wishes for the authors to revise the manuscript.

Hsiao-Hsuan Wang
Ecological Systems Laboratory
Department of Ecology and Conservation Biology
Texas A&M University

Decision letter (RSOS-211380.R0)

Dear Dr Tardy

The Editors assigned to your paper RSOS-211380 "Ecological drivers of tick-borne pathogen infection risk: Inference from a mechanistic modelling approach of host dispersal" have made a decision based on their reading of the paper and any comments received from reviewers.

Regrettably, in view of the reports received, the manuscript has been rejected in its current form. However, a new manuscript may be submitted which takes into consideration these comments.

We invite you to respond to the comments supplied below and prepare a resubmission of your manuscript. Below the referees' and Editors' comments (where applicable) we provide additional requirements. We provide guidance below to help you prepare your revision.

Please note that resubmitting your manuscript does not guarantee eventual acceptance, and we do not generally allow multiple rounds of revision and resubmission, so we urge you to make

every effort to fully address all of the comments at this stage. If deemed necessary by the Editors, your manuscript will be sent back to one or more of the original reviewers for assessment. If the original reviewers are not available, we may invite new reviewers.

Please resubmit your revised manuscript and required files (see below) no later than 21-Apr-2022. Note: the ScholarOne system will 'lock' if resubmission is attempted on or after this deadline. If you do not think you will be able to meet this deadline, please contact the editorial office immediately.

Please note article processing charges apply to papers accepted for publication in Royal Society Open Science (<https://royalsocietypublishing.org/rsos/charges>). Charges will also apply to papers transferred to the journal from other Royal Society Publishing journals, as well as papers submitted as part of our collaboration with the Royal Society of Chemistry (<https://royalsocietypublishing.org/rsos/chemistry>). Fee waivers are available but must be requested when you submit your manuscript (<https://royalsocietypublishing.org/rsos/waivers>).

Thank you for submitting your manuscript to Royal Society Open Science and we look forward to receiving your resubmission. If you have any questions at all, please do not hesitate to get in touch.

on behalf of Dr Christie Bahlai (Associate Editor) and Pete Smith (Subject Editor)
openscience@royalsociety.org

Associate Editor Comments to Author (Dr Christie Bahlai):
Comments to the Author:

We have now received two reviews. Both reviewers agree that the paper presents an interesting subject, but both have also raised some fairly substantial concerns about the framing of the manuscript. I feel there's a good opportunity here to be more explicit about how this work builds on the previous work, but in its current state, it's been flagged by both our editorial infrastructure and one reviewer as being extremely similar to a previous publication. Both reviewers also raised concerns about the modelled results being 'oversold' so in a revision, I would expect to see extrapolations couched in more cautious language.

Reviewer comments to Author:

Reviewer: 1

Comments to the Author(s)

This is an interesting paper exploring the role of host dispersal, landscape characteristics, and pathogen infection process impact tick-borne pathogens in modelled landscapes. Appendices are very clear and well documented. Data and code and supporting documentation are clear and well organized.

However, the authors recently published a paper with a similar title ("Mechanistic movement models reveal ecological drivers of tick-borne pathogen spread" versus this one "Ecological drivers of tick-borne pathogen infection risk: Inference from a mechanistic modelling approach of host dispersal"). They also discuss similar themes in the abstract and intro in both motivations, analytical approaches and some of the results in terms of figures made. Yet it is thus surprising they do not engage with their own work in the discussion. It's critical that the authors make clear

- in all aspects of the paper - what the unique contributions of this follow up paper are and how the findings are both different and also contributing information that the previous paper did not. Additionally, the introduction sets up particular ideas about what will be included in the model and why - but the documentation as is in the methods and then discussion could make clearer for readers what mechanisms are potentially at play and how different assumptions could influence the findings. The authors make bold claims about what their results demonstrate - but it is still a model. More discussion of how the findings fit within empirical field research would be useful, as well as introducing more of the specifics of the system in the introduction and methods, as they greatly shape the model and study design, rather than the end of the discussion.

Line comments

Abstract

20 - "worrying rate" is not specific - what does this mean and to whom? Clearer alternatives could be as simple as "sharply increasing"

21 - be clear from the get go whether public health means people only, 'one health' model of people, ecosystems, and animal health, etc.

25 - does 'host' here mean non-human animal hosts? Or including people?

Introduction

54 - "should" - would be clearer to say that "this pattern implies a small number of hosts are maintaining tick populations and thus contribute..."

56 - be clearer - are you discussing only animal hosts here? Or people too? Because becomes important when you then list the individual characteristics

66 - 67 - suggest re-organizing the paragraph or starting with a different topic sentence. Jumps from dispersal to landscape connectivity. Need to make this paragraph flow of ideas connect more to each other for the reader. The next paragraph discussing habitat clumping, availability and quality, but the importance of those - specifically those elements of landscapes and for tick systems - is not introduced in the introduction before this

67 "animal species invasion" - unclear. Are you referring to predicting invasive animal species' establishment or movement into new areas? Or just discussing movement/establishment generally and not specifically invasions?

72 - 74 - says studies don't do X, but does not cite anything to demonstrate that (from a paper discussing needed research, for example) or providing examples of what such studies do include
74 - 75 = there are other differences between individuals that may explain their movement in the landscape that are not just about "boldness" or fear in movement ecology - why focus on that here?

78 - 83 = no hypotheses? The introduction seems to indicate that you believe you will find certain patterns, based on mechanisms discussed

Methods

89 - 90 = "host dispersal" - is this being used interchangeably with behavior? Questioning distance as measure of boldness? Why? Are there other studies or systems that do this?

107 - 113 = sources for tick natural history information?

141 - 147 = great to be clear about what and was not considered. Can you please add a bit more to explain how this realistically reflects life history / life cycles and provide citations?

162 - please cite software and then cite the individual packages; they have distinct authors and citations

162 - 164 - please explain more about this closed host population and temporal assumptions - what kind of organism(s) are you imagining that this would be appropriate for?

166 - 168 = spring migration where? Transported where?

190 - why transformed?

Results

Figure 3 - consider larger fonts for axes

Discussion

296 - 298 - what makes this model difference from the model in your previous, related paper?

299 - 303 - how do these findings contribute new information in ways that your recent, related paper did not?

306 - 307 = how do you think that your model parameters about tick detachment shape your conclusions? How may they change if you changed those parameters?

311 - how does host grooming show up in your model?

315 - discussion of common hosts - especially if model was made with particular terrestrial hosts in mind - should be in the intro and methods, not introduced in the discussion

319 -

331 - here you discuss the role of climate in questing, not boldness/personality. This seems different than what the introduction was going to explore - in terms of understanding individual differences in movement. Can you say more in the discussion to explain how you believe all of these factors are at play? Can you separate them based on your design?

345 - 349 - discussion of the importance of migratory birds should come up in the intro and methods, since it shaped model and assumption development, not just the discussion

356 - 359 - so is your spatial model 'set' in Canada in terms of seasonal assumptions, etc? That is not discussed in the methods or intro, would be useful to understand this context

361 - this section is named 'heterogeneity in tick burden among hosts' but the 362 - 376 is discussing landscape dynamics, consider more explicit discussion of landscape characteristics as a section in both the results and discussion, especially since it's highlighted in the introduction as motivating the study question and design. Then critical to explore how this compares to existing empirical studies with different landscape contexts - and to reflect on how different parameters would shape the results found

362 - 369 - these results are modeled - how do they compare to empirical studies in this system?

Reviewer: 2

Comments to the Author(s)

This is an interesting, basically well-written paper, which merits publication. However, I am concerned that the authors have "oversold" their model regarding the context within which the current results provide useful information. I feel uncomfortable with the statements like one on lines 390-391: "Importantly, the application of such models (as ours) to tick control strategies could greatly improve planning public health interventions." Admittedly, in the preceding lines (378-386) the authors point out important processes they have chosen to omit from the present model, all of which seem like reasonable simplifications. They then claim their model could easily be extended to include multiple species and real landscape complexity to test various hypotheses regarding tick-borne pathogen infection dynamics (lines 386-389). This also is fine, and I look forward to reading about the results of such model extensions and the associated sensitivity analyses. But the leap from results of such analyses to planning on-the-ground public health interventions is a big one. To quote the often-quoted Science article by Oreskes et al. (1994): "In areas where public policy and public safety are at stake, the burden is on the modeler to demonstrate the degree of correspondence between the model and the material world it seeks to represent and to delineate the limits of that correspondence."

Another concern regarding the presentation/discussion of model results involves statements such as that on lines 307-310: "We found that high probabilities of pathogen transmission from an infectious host to an uninfected larva and low density-dependent mortality rates of larvae attached to hosts resulted in large infected larva burdens per host." This result, per se, can be determined logically without looking at model results. Likewise, low probability of pathogen transmission from an infectious host to an uninfected larva and high density-dependent mortality rates of larvae attached to hosts logically results in relatively small infected larva burdens per host. Results from previous studies (lines 311-318) suggest that real-world host species differ regarding their likelihood of infecting ticks and regarding on-host tick survival rates. Thus,

within this context, it seems that a legitimate question to ask is: “what new knowledge is embodied in these model results?” Or, to paraphrase lines 32-35 as a question: “How (specifically) does the model’s theoretical mechanistic framework provide a better understanding of the tick-borne pathogen infection risk and how (specifically) can it serve as the foundation for applied studies of scenario analyses on public health intervention strategies?” It seems to me that the authors are obliged to provide a clear and direct answer to that question. It also seems to me that assessing the relative influence of the various processes involved in tick-borne pathogen infection dynamics based on a hypothetical “composite” host species, although an interesting and appropriate model evaluation exercise, has limited utility in terms of scenario analyses of public health intervention strategies.

Of much more utility would be an exploration of threshold levels of host-to-tick pathogen transmission rates, on-host and off-host tick survival rates, and host encounter rates that would sustain dangerously high pathogen infection risks to humans. As I understand the model, it certainly has the capability to support such an exploration. For example, given the current parameterization of the model, how high do host encounter rates need to be to sustain dangerously high infection risk (measured as nymphal infection prevalence or some other appropriate metric)? And given that threshold level, what might be feasible habitat modifications or host population control interventions that could maintain host encounter rates below that threshold? Perhaps I am being overly critical, but it is only because I see great unrealized potential regarding the use of the present model. If running and analyzing more simulations is beyond current possibilities, it would be useful to provide at least a description of what the type of model application I suggest might look like within the context of assessing possible public health intervention options.

I have looked at the model equations presented in the Supplementary Material but I have not looked at the model code. The representation of some key processes is phenomenological (e.g., tick questing activity as a function of day of year) and the representation of other key processes is theoretical (e.g., the representation of host dispersal using a non-linear density-dependent emigration function). How one defines “mechanistic modelling,” or, more generally, what one perceives as a cause-effect relationship, obviously is subjective. But I wonder if the use of descriptors like “mechanistic modeling approach” and “theoretical mechanistic framework” serves to clarify or confuse. It seems to me that “spatially-explicit and agent-based,” without sprinkling in the additional descriptors, conveys more clearly the type of model that is presented in the paper.

Just a couple of other items caught my eye. The term “supershedding events” is mentioned in the Abstract and a couple of times in the Discussion, but not in the Results. It would be useful to link that term directly to the results. Implicitly, hosts with high tick burdens potentially re supershedders. But which results indicate the potential presence of supershedders? The only presentation of tick burdens in terms of ticks per host that I can find is in Figure 2, but I cannot relate those partial dependency plots to likelihood of supershedding events without additional guidance. Finally, the journal name is missing on line 521 and on line 527.

My best wishes for the authors to revise the manuscript.

Hsiao-Hsuan Wang
Ecological Systems Laboratory
Department of Ecology and Conservation Biology
Texas A&M University

===PREPARING YOUR MANUSCRIPT===

one version identifying all the changes that have been made (for instance, in coloured highlight, in bold text, or tracked changes);
 a 'clean' version of the new manuscript that incorporates the changes made, but does not highlight them. This version will be used for typesetting if your manuscript is accepted.

===PREPARING YOUR REVISION IN SCHOLARONE===

- Any electronic supplementary material (ESM).
- If you are requesting a discretionary waiver for the article processing charge, the waiver form must be included at this step.
- If you are providing image files for potential cover images, please upload these at this step, and inform the editorial office you have done so. You must hold the copyright to any image provided.
- A copy of your point-by-point response to referees and Editors. This will expedite the preparation of your proof.

- Ensure that your data access statement meets the requirements at <https://royalsociety.org/journals/authors/author-guidelines/#data>. You should ensure that you cite the dataset in your reference list. If you have deposited data etc in the Dryad repository, please include both the 'For publication' link and 'For review' link at this stage.
- If you are requesting an article processing charge waiver, you must select the relevant waiver option (if requesting a discretionary waiver, the form should have been uploaded at Step 3 'File upload' above).
- If you have uploaded ESM files, please ensure you follow the guidance at <https://royalsociety.org/journals/authors/author-guidelines/#supplementary-material> to include a suitable title and informative caption. An example of appropriate titling and captioning may be found at https://figshare.com/articles/Table_S2_from_Is_there_a_trade-off_between_peak_performance_and_performance_breadth_across_temperatures_for_aerobic_scope_in_teleost_fishes_/3843624.

Author's Response to Decision Letter for (RSOS-211380.R0)

See Appendix A.

Decision letter (RSOS-220245.R0)

Dear Dr Tardy,

I am pleased to inform you that your manuscript entitled "Context-dependent host dispersal and habitat fragmentation determine heterogeneity in infected tick burdens: An agent-based modelling study" is now accepted for publication in Royal Society Open Science.

Please remember to make your Dryad dataset 'live' prior to publication, and update any links as needed when you receive a proof to check - you should amend the private 'for review' URL

to a publicly accessible 'for publication' URL. It is good practice to also add data sets, code and other digital materials to your reference list.

on behalf of Dr Christie Bahlai (Associate Editor) and Pete Smith (Subject Editor)
openscience@royalsociety.org

Associate Editor Comments to Author (Dr Christie Bahlai):

The authors have done a very nice job in addressing reviewer comments and polishing the manuscript. I have no further comments.

Appendix A

GRUPE DE RECHERCHE EN
ÉPIDÉMIOLOGIE DES ZOOSES
ET SANTÉ PUBLIQUE

Université de Montréal

February 25th, 2022

Dr. Christie Bahlai
Associate Editor, Royal Society Open Science

OBJECT: Revision of the Manuscript # RSOS-211380

Dear Dr. Bahlai

Thank you very much for sending our manuscript “*Context-dependent host dispersal and habitat fragmentation determine heterogeneity in infected tick burdens: An agent-based modelling study*” to reviewers. We have now integrated most of their comments and submit a revised version of the manuscript. The constructive comments and suggestions of two reviewers significantly improved the clarity and scope of the manuscript, and we hope that you share this opinion.

We also hope that it is now clear that this new article focuses on modelling the effects of animal behaviour on dispersal of ticks and tick-borne pathogens at a local geographic scale using an agent-based modelling approach. Our previous article focused on tick dispersal mechanisms at the continental scale and identified, using a “reaction-advection-diffusion” type model based on partial differential equations, the possibly important, but understudied, role of host dispersal as a determinant of tick-borne disease spread. The current paper therefore takes the next step in this scientific investigation by focusing exclusively on the process of local dispersal of ticks to study the impacts of behaviour of host individuals on fine-scale tick-borne disease dynamics.

Below, we provide a detailed response in blue to each of the comments and we indicate how our manuscript has been revised accordingly.

We hope that you will find our revised version of the manuscript suitable for publication in Royal Society Open Science.

I thank you for reconsidering our revised manuscript and I look forward to hearing from you.

Sincerely yours,

Dr. Olivia Tardy

Postdoctoral Fellow

Research Group on Epidemiology of Zoonoses and Public Health

Faculty of Veterinary Medicine, Université de Montréal,

3200 rue Sicotte, Saint-Hyacinthe (Quebec), J2S 2M2, Canada

E-mail: olivia.tardy@umontreal.ca

Associate Editor Comments to Author (Dr Christie Bahlai):

Comments to the Author:

Comment: We have now received two reviews. Both reviewers agree that the paper presents an interesting subject, but both have also raised some fairly substantial concerns about the framing of the manuscript. I feel there's a good opportunity here to be more explicit about how this work builds on the previous work, but in its current state, it's been flagged by both our editorial infrastructure and one reviewer as being extremely similar to a previous publication. Both reviewers also raised concerns about the modelled results being 'oversold' so in a revision, I would expect to see extrapolations couched in more cautious language.

Response: We thank you and the two reviewers for your time and constructive comments, as well as for the opportunity to revise our manuscript. In the revised version, we now discuss differences between the present work and that of our previous paper (L379-L391). The two models that we built are very different from each other and have been used to respond to different research questions. Specifically, our previous article identified, using a “reaction-advection-diffusion” type model based on partial differential equations, the possibly important, but understudied, role of host dispersal as a determinant of tick-borne disease spread at the continental scale (alongside other important determinants such as tick population dynamics). In this new study, we focus on modelling the effects of animal behaviour on dispersal of ticks and tick-borne pathogens at a local geographic scale using an agent-based modelling approach. We hope that these distinctions, together with the novelty and robustness of the work, are better highlighted in this version. We have also modified the paper title as follows: “Context-dependent host dispersal and habitat fragmentation determine heterogeneity in infected tick burdens: An agent-based modelling study”. Finally, we have added more details on the usefulness of the results and applicability of the model for controlling tick-borne diseases (see the subsection “Implication for controlling tick-borne diseases” of the Discussion section).

Reviewer comments to Author:

Reviewer #1:

Comments to the Author(s):

This is an interesting paper exploring the role of host dispersal, landscape characteristics, and pathogen infection process impact tick-borne pathogens in modelled landscapes. Appendices are very clear and well documented. Data and code and supporting documentation are clear and well organized. However, the authors recently published a paper with a similar title (“Mechanistic movement models reveal ecological drivers of tick-borne pathogen spread” versus this one “Ecological drivers of tick-borne pathogen infection risk: Inference from a mechanistic modelling approach of host dispersal”). They also discuss similar themes in the abstract and intro in both motivations, analytical approaches and some of the results in terms of figures made. Yet it is thus surprising they do not engage with their own work in the discussion. It's critical that the authors make clear – in all aspects of the paper – what the unique contributions of this follow up paper are and how the findings are both different and also contributing information that the previous paper did not. Additionally, the introduction sets up particular ideas about what will be

included in the model and why – but the documentation as is in the methods and then discussion could make clearer for readers what mechanisms are potentially at play and how different assumptions could influence the findings. The authors make bold claims about what their results demonstrate – but it is still a model. More discussion of how the findings fit within empirical field research would be useful, as well as introducing more of the specifics of the system in the introduction and methods, as they greatly shape the model and study design, rather than the end of the discussion.

Line comments:

ABSTRACT

Comment 1: L20 – “worrying rate” is not specific – what does this mean and to whom? Clearer alternatives could be as simple as “sharply increasing”

Response: Thank you for pointing this out. “worrying rate” means that the number of reported tick-borne disease cases among humans continues to dramatically increase over the past decade. For example, the number of human Lyme disease cases reported in Canada increased from 144 in 2009 to 917 in 2015 [1]. Following your suggestion, the sentence has been revised as follows: “As the incidence of tick-borne diseases has sharply increased over the past decade, with serious consequences for human and animal health, there is a need to identify ecological drivers contributing to heterogeneity in tick-borne disease risk.”.

Comment 2: L21 – be clear from the get go whether public health means people only, ‘one health’ model of people, ecosystems, and animal health, etc.

Response: Thank you for the comment. We have replaced “public health” with “human and animal health”.

Comment 3: L25 – does ‘host’ here mean non-human animal hosts? Or including people?

Response: Thank you for the question. Indeed, “host” means non-human animal hosts. We now specify that hosts refer to animal hosts (L24).

INTRODUCTION

Comment 4: L54 – “should” – would be clearer to say that “this pattern implies a small number of hosts are maintaining tick populations and thus contribute...”

Response: Following your suggestion, the sentence has been revised as follows: “This pattern implies that a small number of animal hosts are maintaining tick populations and thus contribute to natural transmission cycles of tick-borne pathogens, given that these hosts are most likely both to become infected and to be the source of infection for many uninfected ticks.”.

Comment 5: L56 – be clearer – are you discussing only animal hosts here? Or people too? Because becomes important when you then list the individual characteristics

Response: Indeed, we are discussing only animal hosts here. We now specify that hosts refer to animal hosts (L55, L56).

Comment 6: L66 – 67 – suggest re-organizing the paragraph or starting with a different topic sentence. Jumps from dispersal to landscape connectivity. Need to make this paragraph flow of ideas connect more to each other for the reader. The next paragraph discussing habitat clumping, availability and quality, but the importance of those – specifically those elements of landscapes and for tick systems - is not introduced in the introduction before this

Response: We have reorganized the paragraph by focusing on the importance of host dispersal for tick systems (L75-L103).

Comment 7: L67 “animal species invasion” – unclear. Are you referring to predicting invasive animal species’ establishment or movement into new areas? Or just discussing movement/establishment generally and not specifically invasions?

Response: Thank you for this comment. We are referring to animal species movement and establishment generally. We have modified the sentence as follows: “It has been recognized that the ecological process of dispersal (i.e. individual movement from a natal habitat patch to a new patch) is essential to understand the expanding distributions of species in response to anthropogenic environmental changes [2-4], including ticks and tick-borne pathogens [5].”.

Comment 8: L72 – 74 – says studies don’t do X, but does not cite anything to demonstrate that (from a paper discussing needed research, for example) or providing examples of what such studies do include

Response: We now add references (including papers that discuss needed research) as follows: “In many spatial models projecting future trends in tick-borne diseases [6-8], host dispersal behaviour is linked to landscape characteristics in simplistic ways (e.g. modelled by a simple parameter) [9]. However, significant advances have been made in understanding and modelling dispersal mechanisms [10-12]. In particular, modelling the three context-dependent phases of the dispersal process, such as emigration, transfer (or interpatch movement) and settlement (or immigration) [9], is important for making reliable projections of spatio-temporal distributions of animal species [12-14]. In addition, inter-individual variability in host dispersal behaviour is often ignored in tick-borne disease models [6-8, 15], most of which treat dispersal as species- or population-level average behaviour [16].”.

Comment 9: L74 – 75 – there are other differences between individuals that may explain their movement in the landscape that are not just about “boldness” or fear in movement ecology – why focus on that here?

Response: We focused on boldness-related host behaviour across a shy-bold spectrum because empirical studies showed that this personality trait can have consequences for tick parasitism [17, 18]. For example, exploratory and bold hosts having larger home ranges and moving further from their natal site are more likely to encounter ticks, have high tick burdens and act as super-spreaders of ticks and tick-borne pathogens [17]. These individuals may have a central role in tick-borne pathogen transmission and spread processes [19], and should be considered in theoretical or applied epidemiological models of tick-borne diseases. We now state this in the text (L87-L95).

Comment 10: L78 – 83= no hypotheses? The introduction seems to indicate that you believe you will find certain patterns, based on mechanisms discussed

Response: We don't state hypotheses in the introduction because we conducted exploratory research by performing a global sensitivity analysis (SA) of our spatially explicit agent-based model (ABM). The objective of the SA was to assess the relative influence of ABM input parameters on infected tick burdens on hosts and infected tick distributions among hosts. The SA allowed the identification of host-intrinsic and -extrinsic parameters that may be those among the most influential on tick-borne disease risk, and those for which research effort should be prioritised to better understand the ecological processes underlying tick-borne disease risk. Indeed, we found that some well-studied parameters were among the most influential predictors, which shows that the model behaves in a biologically plausible way. We now state this in the text (L109-L111, L236-L239).

METHODS

Comment 11: L89 – 90 = “host dispersal” – is this being used interchangeably with behavior? Questioning distance as measure of boldness? Why? Are there other studies or systems that do this?

Response: Indeed, “host dispersal” is being used interchangeably with behaviour. Host dispersal behaviour is a key component of our model where the three dispersal phases (i.e. emigration, transfer and settlement) interact dynamically with the environment. In addition, our model incorporates inter-individual variability and context dependencies (i.e. conspecific density and availability of habitat suitable for hosts and ticks) in the dispersal process. Similarly to Palmer et al. [16], boldness was measured by a cost adjustment parameter, which was applied to the movement cost values of each cell ($C_{\text{Land cover}}$). The parameter gives an adjusted cost for the cell of $C_{\text{Land cover}}^j$ that can either expand ($j < 1$) or reduce ($j > 1$) movement ability of host individuals across the landscape, which represents situations where individuals are more bold or shy than an “reference” individual ($j = 1$) [16] (L294-L298 in Appendix A). In the context of tick-borne diseases, there are no models that incorporate the complexity of boldness-related host behaviour.

Comment 12: L107 – 113 = sources for tick natural history information?

Response: Thank you for this observation. We have added references and we have mentioned the tick family (i.e. ixodid or hard-bodied ticks) (L148).

Comment 13: L141 – 147 = great to be clear about what and was not considered. Can you please add a bit more to explain how this realistically reflects life history / life cycles and provide citations?

Response: Thank you for this comment. The demographic turnover process in our spatially explicit agent-based model is a simplified representation of the individual recruitment process observed in reality. We assumed that birth and death rates are equal and vary over time. It has been shown that rodent populations can experience boom-bust cycles of growth in response to episodic production of acorns in locations where oak trees are abundant [20]. In our study, we did not simulate seasonally or inter-annually fluctuating resources, but instead, we ran simulations across spatially heterogeneous landscapes varying in clumping and availability of suitable habitat. Further studies are needed to explore the effects of resource pulses. As in other studies (e.g. [21]), we assumed that host populations are demographically stable given that wildlife host fitness (i.e. survival and reproduction) is not significantly affected, to our current knowledge, by high tick burdens [22] and the bacterium *B. burgdorferi* [23]. We now state this in the section “Host population dynamics” of the Materials and Methods section.

Comment 14: L162 – please cite software and then cite the individual packages; they have distinct authors and citations

Response: Thank you for this observation. We now cite the software.

Comment 15: L162 – 164 – please explain more about this closed host population and temporal assumptions – what kind of organism(s) are you imagining that this would be appropriate for?

Response: Thank you for this comment. Our spatially explicit agent-based model (ABM) runs in a cell-based setting [24] in which the ecological processes associated with host dispersal behaviour and host population dynamics act at the cell scale. Consequently, the context dependencies operating in the dispersal process happen at this scale. Host individuals that are located in a given cell define a distinct open population (i.e. experiencing recruitment through births and immigrations, or losses through deaths and emigrations). However, the host populations are geographically closed because there is no recruitment or loss of individuals from outside the study area. We simulated hypothetical host species that represent avian and terrestrial animal species having different individual densities, dispersal abilities and lifespans. The polygon-based version of the ABM in which the ecological processes act at the polygon scale (i.e. a polygon being defined by an assemblage of landscape cells) can be applied to real landscapes in order to simulate a broad range of real organisms (e.g. invertebrates, mammals, birds) [25]. We now state this in the Materials and Methods section (L123-L128, L133-L135, L220-L222).

Comment 16: L166 – 168 = spring migration where? Transported where?

Response: We have revised the sentence to clarify this point as follows: “At each year, uninfected and infected ticks were transported by migratory passerine birds into the landscape cells during the spring migration period (May 01 – June 30). We did not explicitly model northward spring migration, but instead, migration provided a supply of ticks to approximate the effects of tick introduction by migratory birds.”.

Comment 17: L190 – why transformed?

Response: Thank you for pointing this out. We have included the information in the text as follows: “The average burdens of infected ticks per host were log-transformed in the BRT models to stabilize the variance and to meet the assumption of Gaussian error distributions [26].”.

RESULTS

Comment 18: Figure 3 – consider larger fonts for axes

Response: We now use larger fonts for axes.

DISCUSSION

Comment 19: L296 – 298 – what makes this model difference from the model in your previous, related paper?

Response: In our previous paper, we built a “reaction-advection-diffusion” type model that simulates northward invasion of ticks by migratory birds and terrestrial hosts over large spatial

scales across spatially heterogeneous landscapes varying in resource aggregation. This population-level model with a continuous representation of space and time integrated a mechanistic formulation of host movement to investigate how ecological predictors associated with the host movement process, the tick and host demographic processes, resource aggregation and the pathogen infection process interact to drive the speed of global spread of infected ticks, tick infection prevalence and infected tick density. The outcome of that work was that long-distance dispersal of migratory birds was likely important in tick-borne disease spread dynamics. The present paper takes the next step in the investigation of the role of host dispersal by focusing exclusively on the process of local dispersal of hosts to study the impacts of behaviour of host individuals on fine-scale tick-borne disease dynamics. To do this, we built a spatially explicit agent-based model (ABM) with a discretized representation of space and time, which allowed modelling fine-scale individual-level movement. The ABM simulates dispersal of ticks by hypothetical avian or terrestrial hosts over fine spatial scales across spatially heterogeneous landscapes varying in clumping and proportion of habitat suitable for hosts and ticks. This model was used to identify ecological drivers of infected tick burdens on hosts and infected tick distributions among hosts. While the two models integrate ecological processes governing both host movement, tick and host population dynamics and pathogen infection, they differ in the way that these processes are modelled considering the model structural properties (i.e. continuous vs. discretized representation of space and time, population-level vs. individual-level model, large-scale vs. fine-scale study), which induces differences in the input parameters that are used to perform the global sensibility analysis (SA). Given that the research objective was different between our two papers, we did not use the same response variables to build boosted regression tree models in the SA. Contrary to our previous paper, we considered distinct life stages of ticks (e.g. larvae, nymphs and adults) in our ABM. We have included these details in the Discussion section (L379-L391).

Comment 20: L299 – 303 – how do these findings contribute new information in ways that your recent, related paper did not?

Response: As mentioned in the response to the comment 19, the model input parameters and the output variables that are used to fit the boosted regression tree models in the global sensitivity analysis are different from our previous paper. Consequently, our present findings contribute to new information. In particular, we found that the type of host settlement strategy and the proportion of habitat suitable for hosts strongly determined heterogeneity in infected tick burdens among hosts. From these results, super-spreaders of infected ticks, i.e. host individuals that have disproportionate burdens of infected ticks, were more likely to be present in highly fragmented landscapes with low levels of habitat suitable for hosts (≤ 0.25) and to have settlement strategies that rely on both finding suitable habitat while avoiding cells with high densities of conspecifics. These findings are of importance for tick-borne disease control because super-spreading of infected engorged larvae has a strong impact on tick-borne pathogen transmission. The empirical evidence or modelling assumptions supporting our results are, to date, absent. We state this in the Discussion section (L397-L400) and the subsection “Infected tick distribution among hosts” of the Discussion section. In addition, our simulations indicated that ecological predictors of infected tick burdens on hosts differed among the post-egg life stages of ticks (i.e. larvae, nymphs and adults), with tick attachment and detachment, tick questing activity and pathogen transmission dynamics identified as key processes, in a way consistent with our knowledge of tick-borne disease ecology. These results demonstrate that the model has the potential to

reproduce patterns observed in real systems. We now state this in the Discussion section (L392-L397).

Comment 21: L306 – 307 = how do you think that your model parameters about tick detachment shape your conclusions? How may they change if you changed those parameters?

Response: We found that the slope at the inflection point of the on-host density-dependent larva mortality function (α_d^{aL}) was the most influential input parameter among the input parameters associated with the tick attachment-detachment process. This predictor controls the density-dependent mortality rate of larvae attached to hosts. The partial dependency plot showed that the slope at the inflection point of the on-host density-dependent larva mortality function decreased nonlinearly with increasing the infected larva burden on hosts. This decrease was rapid when the slope was low (≤ 0.02), and became increasingly slow as the slope increased (Figure 2a). As expected, the density-dependent mortality rate of larvae feeding on hosts decreased with increasing the infected larva burden. An increase in tick burden can induce a reduction in feeding success due to an increase in host grooming behaviour [27] or acquired immunological resistance of hosts [28], while a decrease in tick burden would reduce grooming behaviour or acquired immunological resistance and thus would favour survival of ticks feeding on hosts [29]. We have included these details in the Results (L287-L292) and Discussion (L409-L414) sections.

Comment 22: L311 – how does host grooming show up in your model?

Response: At each time step, we applied density-dependent mortality rates to ticks feeding on hosts to represent density-dependent mortality that could be due to host grooming behaviour and the effects of acquired resistance to ticks – both of which are dependent on density [30, 31]. The equations that are used to calculate the density-dependent mortality rate for each post-egg life stage of ticks (i.e. larvae, nymphs and adults) are mentioned in Appendix B.

Comment 23: L315 – discussion of common hosts – especially if model was made with particular terrestrial hosts in mind – should be in the intro and methods, not introduced in the discussion

Response: The model has not been built with particular hosts in mind. We simulated hypothetical hosts that represent a broad range of avian and terrestrial animal species with realistic characteristics, including dispersal ability, individual density and lifespan (L133-L135). In the Discussion section, we have cited some examples of host species to illustrate our results.

Comment 24: L319 – 331 – here you discuss the role of climate in questing, not boldness/personality. This seems different than what the introduction was going to explore – in terms of understanding individual differences in movement. Can you say more in the discussion to explain how you believe all of these factors are at play? Can you separate them based on your design?

Response: We found that the base host-finding rate of questing nymphs and the mortality rate of ticks developing from engorged larvae into questing nymphs in breeding habitats were the most influential input parameters predicting the infected nymph burden on hosts. We thus described in the Discussion section how these parameters affect the infected nymph burden. For example, the mortality rate of ticks developing from engorged larvae into questing nymphs in breeding habitats controls the questing nymph populations in the landscape by affecting the proportion of engorged larvae that moult into questing nymphs during the interstadial development phase. This

lines up with field studies showing that high tick burdens on small-mammal hosts coincide with high densities of host-seeking ticks on vegetation [32, 33]. Tick mortality during the interstadial development phase is largely influenced by temperature and humidity (reviewed in [34]), with extremes of temperature, drowning and desiccation being the main determinants of tick mortality. Climate also determines tick questing activity (reviewed in [35]) and thus impacts on encounter rates between hosts and ticks, which can experience different mortality rates according to temperature conditions. We state this in the text (L425-L434). We did not integrate input parameters related to climate in our model and it is for this reason that we did not mention the role of climate in the Introduction section. Therefore, we cannot separate the effects of climate based on our design. To avoid confusion, we have added the following sentence: “We did not integrate input parameters related to climate in the ABM, although temperature and humidity would affect tick burdens on hosts because of their direct effects on tick activity and survival [35].”. The contribution of our study lies in the construction of a model that incorporates a mechanistic formulation of host dispersal behaviour and accounts for personality-dependent dispersal behavioural variations (i.e. boldness and shyness) in a tick-borne disease context. This is the reason why we discussed individual differences in dispersal behaviour in the Introduction section. However, the results of our global sensitivity analysis suggest that host personality traits are less important than the tick questing activity and interstadial development phases in predicting infected nymph burdens on hosts.

Comment 25: L345 – 349 – discussion of the importance of migratory birds should come up in the intro and methods, since it shaped model and assumption development, not just the discussion

Response: We have added the importance of migratory birds in the Materials and Methods section as follows: “In addition, migratory birds can transport ticks and pathogens over long distances to new non-endemic areas, and can thus contribute to range expansion of both ticks and tick-borne pathogens [36-38]. For example, migratory passerine birds have facilitated northward range expansion of *I. scapularis* ticks and the bacterial cause of Lyme disease, *B. burgdorferi*, in North America [39-41] during their spring migration, by virtue of their role as tick hosts and reservoir hosts for tick-borne pathogens.”.

Comment 26: L356 – 359 – so is your spatial model ‘set’ in Canada in terms of seasonal assumptions, etc? That is not discussed in the methods or intro, would be useful to understand this context

Response: Our model is not specific to a particular study area (e.g. in Canada), but can be adapted for any regions or species with sufficient empirical data as a wide set of input parameters are integrated in the model. For example, the phenomenological equations are flexible enough to model any patterns of changes in seasonal dynamics of questing tick populations. We have clarified this in the manuscript (L135-L137).

Comment 27: L361 – this section is named ‘heterogeneity in tick burden among hosts’ but the 362 – 376 is discussing landscape dynamics, consider more explicit discussion of landscape characteristics as a section in both the results and discussion, especially since it’s highlighted in the introduction as motivating the study question and design. Then critical to explore how this compares to existing empirical studies with different landscape contexts – and to reflect on how different parameters would shape the results found

Response: It is especially the investigation of the effects of host dispersal behaviour on infected tick burdens and infected tick distributions among hosts that motivated the study question and design. We have thus reorganized the Introduction section by focusing on the importance of host dispersal rather than landscape characteristics for tick systems (L75-L103) and we have compared our results to empirical or modelling studies.

Comment 28: L362 – 369 – these results are modeled – how do they compare to empirical studies in this system?

Response: We found that the type of host settlement strategy (i.e. individual's decisions to stop in a given habitat cell) and the proportion of habitat suitable for hosts strongly determined heterogeneity in infected tick burdens among hosts. From our results, super-spreaders of infected ticks were more likely to be present in highly fragmented landscapes with low levels of habitat suitable for hosts (≤ 0.25) and to have settlement strategies that rely on both finding suitable habitat while avoiding cells with high densities of conspecifics. The empirical evidence or modelling assumptions supporting these results are, to date, absent. On the basis of other studies, we consider that the explanation of observed patterns is related to the effects of context-dependent settlement behaviour and landscape characteristics on the rate of host spread [10, 42]. For example, the modelling study of Bocedi et al. [10] revealed that settlement strategies involving either only finding suitable habitat or finding suitable habitat with negative density dependence induced higher population spread rates compared to density-independent settlement strategies, with higher differences observed in landscapes with low rather than high amount of suitable habitat. These results suggest the existence of a “shadow effect”, a phenomenon that happens when suitable habitat patches intercept individuals dispersing through the matrix, compelling them to stop in cells close to their natal cell, and thus decrease the probability of dispersing towards other patches, with the consequence of decreasing the spread rates [43]. We state this in the subsection “Infected tick distribution among hosts” of the Discussion section.

Reviewer #2:

Comments to the Author(s)

Comment 1: This is an interesting, basically well-written paper, which merits publication. However, I am concerned that the authors have “oversold” their model regarding the context within which the current results provide useful information. I feel uncomfortable with the statements like one on lines 390-391: “Importantly, the application of such models (as ours) to tick control strategies could greatly improve planning public health interventions.” Admittedly, in the preceding lines (378-386) the authors point out important processes they have chosen to omit from the present model, all of which seem like reasonable simplifications. They then claim their model could easily be extended to include multiple species and real landscape complexity to test various hypotheses regarding tick-borne pathogen infection dynamics (lines 386-389). This also is fine, and I look forward to reading about the results of such model extensions and the associated sensitivity analyses. But the leap from results of such analyses to planning on-the-ground public health interventions is a big one. To quote the often-quoted Science article by Oreskes et al. (1994): “In areas where public policy and public safety are at stake, the burden is on the modeler to demonstrate the degree of

correspondence between the model and the material world it seeks to represent and to delineate the limits of that correspondence.”

Response: Identifying ecological drivers of heterogeneity in infected tick burden patterns (also called “super-spreading” of infected ticks) is important to design effective control measures aiming at reducing human and animal exposure to infected tick bites [44]. The efficiency of tick-borne disease control could thus be improved by identifying super-spreaders of infected ticks. The relative contribution of different ecological factors in predicting infected tick distribution patterns among hosts should be evaluated to characterize super-spreading of infected ticks. However, super-spreading of ticks and tick-borne pathogens have received limited theoretical and empirical attention in the context of tick-borne diseases (but see [32, 44]). We developed a spatially explicit agent-based model as a first step in this research line. Our findings suggest that host individuals using settlement strategies with negative density dependence (i.e. individuals that are less likely to settle a habitat cell containing high density of conspecifics) and thus exhibiting active dispersal would have more chance of acting as super-spreaders of infected ticks. We also found that highly fragmented landscapes with a small proportion of habitat suitable for hosts (≤ 0.25) would be more likely to promote super-spreading of infected ticks. These findings are of importance for tick-borne disease control because super-spreading of infected engorged larvae has a strong impact on tick-borne pathogen transmission. However, further work that extends the ABM is required for testing the efficiency of control methods targeting super-spreaders of infected ticks. We now state this in the section “Implication for controlling tick-borne diseases” of the discussion.

Comment 2: Another concern regarding the presentation/discussion of model results involves statements such as that on lines 307-310: “We found that high probabilities of pathogen transmission from an infectious host to an uninfected larva and low density-dependent mortality rates of larvae attached to hosts resulted in large infected larva burdens per host.” This result, per se, can be determined logically without looking at model results. Likewise, low probability of pathogen transmission from an infectious host to an uninfected larva and high density-dependent mortality rates of larvae attached to hosts logically results in relatively small infected larva burdens per host. Results from previous studies (lines 311-318) suggest that real-world host species differ regarding their likelihood of infecting ticks and regarding on-host tick survival rates. Thus, within this context, it seems that a legitimate question to ask is: “what new knowledge is embodied in these model results?” Or, to paraphrase lines 32-35 as a question: “How (specifically) does the model’s theoretical mechanistic framework provide a better understanding of the tick-borne pathogen infection risk and how (specifically) can it serve as the foundation for applied studies of scenario analyses on public health intervention strategies?” It seems to me that the authors are obliged to provide a clear and direct answer to that question. It also seems to me that assessing the relative influence of the various processes involved in tick-borne pathogen infection dynamics based on a hypothetical “composite” host species, although an interesting and appropriate model evaluation exercise, has limited utility in terms of scenario analyses of public health intervention strategies.

Response: Thank you for pointing this out. It has been recognized that the ecological process of dispersal (i.e. individual movement from a natal habitat patch to a new patch) is essential to understand the expanding distributions of species in response to anthropogenic environmental changes [2-4], including ticks and tick-borne pathogens [5]. In many spatial models projecting future trends in tick-borne diseases [6-8], host dispersal behaviour is linked to landscape

characteristics in simplistic ways (e.g. modelled by a simple parameter) [9]. However, significant advances have been made in understanding and modelling dispersal mechanisms [10-12]. In particular, modelling the three context-dependent phases of the dispersal process, such as emigration, transfer (or interpatch movement) and settlement (or immigration) [9], is important for making reliable projections of spatio-temporal distributions of animal species [12-14]. In addition, inter-individual variability in host dispersal behaviour is often ignored in tick-borne disease models [6-8, 15], most of which treat dispersal as species- or population-level average behaviour [16]. However, empirical studies showed that personality-related host behaviour (i.e. consistent individual differences in behaviour over time or from contexts [45]) can induce changes in a large range of ecological traits (e.g. habitat use, dispersal, fitness) [46-48], which can have consequences for tick parasitism [17, 18]. To properly explore host dispersal, we need to develop spatially explicit and individual-based movement mechanistic models under the general conceptual framework of Nathan et al. [49]. The use of such models that integrate greater realism in the dispersal process should lead to a better understanding of tick-host-pathogen interactions. Our study contributes to this line of research. In particular, we found that the type of host settlement strategy and the proportion of habitat suitable for hosts strongly determined heterogeneity in infected tick burdens among hosts. From our results, super-spreaders of infected ticks were more likely to be present in highly fragmented landscapes with low levels of habitat suitable for hosts (≤ 0.25) and to have settlement strategies that rely on both finding suitable habitat while avoiding cells with high densities of conspecifics. The empirical evidence or modelling assumptions supporting these results are, to date, absent. These findings highlight the importance of context-dependent host dispersal and habitat fragmentation in super-spreading of infected ticks. Our simulations also showed that ecological predictors of infected tick burdens on hosts differed among the post-egg life stages of ticks (i.e. larvae, nymphs and adults), with tick attachment and detachment, tick questing activity and pathogen transmission dynamics identified as key processes, in a way consistent with our knowledge of tick-borne disease ecology. These results demonstrate that the ABM has the potential to reproduce patterns observed in real systems. Finally, movement mechanistic models should provide a useful tool to prioritise tick and tick-borne pathogen control actions by offering the possibility for managers to test different host- and/or environment-targeted control strategies and to identify what strategies would be effective in reducing tick-borne disease risk. We now state this in the Introduction (L75-L103) and Discussion (L392-L400) sections, and the subsection “Infected tick distribution among hosts” of the Discussion section.

Comment 3: Of much more utility would be an exploration of threshold levels of host-to-tick pathogen transmission rates, on-host and off-host tick survival rates, and host encounter rates that would sustain dangerously high pathogen infection risks to humans. As I understand the model, it certainly has the capability to support such an exploration. For example, given the current parameterization of the model, how high do host encounter rates need to be to sustain dangerously high infection risk (measured as nymphal infection prevalence or some other appropriate metric)? And given that threshold level, what might be feasible habitat modifications or host population control interventions that could maintain host encounter rates below that threshold? Perhaps I am being overly critical, but it is only because I see great unrealized potential regarding the use of the present model. If running and analyzing more simulations is beyond current possibilities, it would be useful to provide at least a description of

what the type of model application I suggest might look like within the context of assessing possible public health intervention options.

Response: Thank you for this comment. Indeed, the model has the capacity to assess threshold levels of input parameters that are able to sustain high infection risk. However, the model would have to be applied in real landscapes to find such thresholds. Now that the model is prepared to work with landscape complexity, combined modelling and empirical studies can take place to provide threshold values and to assess the impacts of public health interventions. The model can be used to test different host- and/or environment-targeted intervention strategies and to identify what strategies would be effective in reducing tick-borne pathogen infection risk for humans. However, two steps should be implemented before assessing potential public health intervention options. As our model incorporates a large number of input parameters ($N = 56$), among which some are poorly documented in the literature, an exploratory sensitivity analysis should be performed in a first step to identify the most influential parameters that need to be prioritised for intervention scenario analyses. In fact, exploring different threshold levels for each of 56 parameters in these scenario analyses would not be a good strategy and of limited interest, as well as computationally expensive in knowing that several replications per parameter set must be conducted due to model stochasticity. It is in this perspective that we performed a global sensitivity analysis in this study. Our results can thus be used for future intervention scenario analyses. In a second step, our model should be extended to more ecologically complex tick-host-pathogen systems by integrating both multiple species and real landscape complexity. For example, rodent and deer populations could be simulated in a first time. It would be simple to perform this model modification if interspecific interactions are not considered in the system. Finally, land cover types in real landscapes should be associated with additional movement costs, which requires the extension of some input parameters that are linked to movement costs (e.g. tick mortality rates during the questing and interstadial development phases in each habitat type). As suggested, we have added threshold levels for the input parameters in the results.

Comment 4: I have looked at the model equations presented in the Supplementary Material but I have not looked at the model code. The representation of some key processes is phenomenological (e.g., tick questing activity as a function of day of year) and the representation of other key processes is theoretical (e.g., the representation of host dispersal using a non-linear density-dependent emigration function). How one defines “mechanistic modelling,” or, more generally, what one perceives as a cause-effect relationship, obviously is subjective. But I wonder if the use of descriptors like “mechanistic modeling approach” and “theoretical mechanistic framework” serves to clarify or confuse. It seems to me that “spatially-explicit and agent-based,” without sprinkling in the additional descriptors, conveys more clearly the type of model that is presented in the paper.

Response: Thank you for this comment. The model presented in this paper is a stochastic and spatially explicit agent-based model (ABM) that includes a mechanistic representation of host dispersal behaviour by explicitly describing how host individuals move in response to landscape characteristics. This representation type allows increasing realism and applicability of the ABM. We added this in the introduction (L104-L109) and as suggested, we have referred to the model in other paper sections as a “spatially explicit agent-based model” without using additional descriptors.

Comment 5: Just a couple of other items caught my eye. The term “supershedding events” is mentioned in the Abstract and a couple of times in the Discussion, but not in the Results. It would be useful to link that term directly to the results. Implicitly, hosts with high tick burdens potentially re supershedders. But which results indicate the potential presence of supershedders? The only presentation of tick burdens in terms of ticks per host that I can find is in Figure 2, but I cannot relate those partial dependency plots to likelihood of supershedding events without additional guidance. Finally, the journal name is missing on line 521 and on line 527.

Response: Thank you for pointing this out. We replaced the “super-shedding” and “super-shedders” terms with the “super-spreading” and “super-spreaders” terms, which appears to be more appropriate for the description of individuals having disproportionately higher levels of tick burdens than the average. We now mention these terms in the results as follows: “This suggests that a settlement strategy with negative density dependence would promote super-spreading of infected ticks (Figure 3). [...], suggesting that hosts are more likely to act as super-spreaders of infected ticks in highly fragmented landscapes with a small proportion of breeding habitat (≤ 0.25). [...]. This suggests that super-spreaders of infected nymphs are more likely to be found in areas where encounter rates with questing nymphs are low.”. We used the Hoover concentration index (0 – 1) as an indicator for the potential presence of super-spreaders of infected ticks. This index takes the value of zero when all hosts have the same infected tick burden and the value of one when all infected ticks are concentrated on a single host. High values for the Hoover concentration index thus reveal the presence of super-spreaders of infected ticks. We have added this sentence in the text (L254-L255). The results of the global sensibility analysis for the Hoover concentration index are represented in Figure 3 and described in the subsection “Infected tick distribution among hosts” of the Results section. Thank you for this observation. We now add the journal name.

My best wishes for the authors to revise the manuscript.

Hsiao-Hsuan Wang
Ecological Systems Laboratory
Department of Ecology and Conservation Biology
Texas A&M University

References

- [1] Gasmi, S., Ogden, N.H., Lindsay, L.R., Burns, S., Fleming, S., Badcock, J., Hanan, S., Gaulin, C., Leblanc, M.A., Russell, C., et al. 2017 Surveillance for Lyme disease in Canada: 2009-2015. *Canada Communicable Disease Report* **43**, 194-199. (doi:10.14745/ccdr.v43i10a01).
- [2] Clobert, J., Baguette, M., Benton, T.G. & Bullock, J.M. 2013 *Dispersal ecology and evolution*. Oxford, UK, Oxford University Press.
- [3] Travis, J.M.J., Delgado, M., Bocedi, G., Baguette, M., Bartoń, K., Bonte, D., Boulangeat, I., Hodgson, J.A., Kubisch, A., Penteriani, V., et al. 2013 Dispersal and species' responses to climate change. *Oikos* **122**, 1532-1540. (doi:10.1111/j.1600-0706.2013.00399.x).
- [4] Kokko, H. & López-Sepulcre, A. 2006 From individual dispersal to species ranges: perspectives for a changing world. *Science* **313**, 789-791. (doi:10.1126/science.1128566).
- [5] Tsao, J.I., Hamer, S.A., Han, S., Sidge, J.L. & Hickling, G.J. 2021 The contribution of wildlife hosts to the rise of ticks and tick-borne diseases in North America. *J. Med. Entomol.* **58**, 1565-1587. (doi:10.1093/jme/tjab047).
- [6] Nadolny, R.M. & Gaff, H.D. 2018 Modelling the effects of habitat and hosts on tick invasions. *Letters in Biomathematics* **5**, 2-29. (doi:10.1080/23737867.2017.1412811).
- [7] Halsey, S.J. & Miller, J.R. 2018 A spatial agent-based model of the disease vector *Ixodes scapularis* to explore host-tick associations. *Ecol. Model.* **387**, 96-106. (doi:10.1016/j.ecolmodel.2018.09.005).
- [8] Tonelli, B.A. & Dearborn, D.C. 2019 An individual-based model for the dispersal of *Ixodes scapularis* by ovenbirds and wood thrushes during fall migration. *Ticks and Tick-Borne Diseases* **10**, 1096-1104. (doi:10.1016/j.ttbdis.2019.05.012).
- [9] Bowler, D.E. & Benton, T.G. 2005 Causes and consequences of animal dispersal strategies: relating individual behaviour to spatial dynamics. *Biological Reviews* **80**, 205-225. (doi:10.1017/S1464793104006645).
- [10] Bocedi, G., Zurell, D., Reineking, B. & Travis, J.M.J. 2014 Mechanistic modelling of animal dispersal offers new insights into range expansion dynamics across fragmented landscapes. *Ecography* **37**, 1240-1253. (doi:10.1111/ecog.01041).
- [11] Clobert, J., Le Galliard, J.-F., Cote, J., Meylan, S. & Massot, M. 2009 Informed dispersal, heterogeneity in animal dispersal syndromes and the dynamics of spatially structured populations. *Ecol. Lett.* **12**, 197-209. (doi:10.1111/j.1461-0248.2008.01267.x).

- [12] Travis, J.M.J., Mustin, K., Bartoń, K.A., Benton, T.G., Clobert, J., Delgado, M.M., Dytham, C., Hovestadt, T., Palmer, S.C.F., Van Dyck, H., et al. 2012 Modelling dispersal: an eco-evolutionary framework incorporating emigration, movement, settlement behaviour and the multiple costs involved. *Methods Ecol Evol* **3**, 628-641. (doi:10.1111/j.2041-210X.2012.00193.x).
- [13] Aben, J., Bocedi, G., Palmer, S.C.F., Pellikka, P., Strubbe, D., Hallmann, C., Travis, J.M.J., Lens, L. & Matthysen, E. 2016 The importance of realistic dispersal models in conservation planning: application of a novel modelling platform to evaluate management scenarios in an Afrotropical biodiversity hotspot. *J. Appl. Ecol.* **53**, 1055-1065. (doi:10.1111/1365-2664.12643).
- [14] Ovenden, T.S., Palmer, S.C.F., Travis, J.M.J. & Healey, J.R. 2019 Improving reintroduction success in large carnivores through individual-based modelling: how to reintroduce Eurasian lynx (*Lynx lynx*) to Scotland. *Biol. Conserv.* **234**, 140-153. (doi:10.1016/j.biocon.2019.03.035).
- [15] Li, S., Hartemink, N., Speybroeck, N. & Vanwambeke, S.O. 2012 Consequences of landscape fragmentation on Lyme disease risk: a cellular automata approach. *PLOS ONE* **7**, e39612. (doi:10.1371/journal.pone.0039612).
- [16] Palmer, S.C.F., Coulon, A. & Travis, J.M.J. 2014 Inter-individual variability in dispersal behaviours impacts connectivity estimates. *Oikos* **123**, 923-932. (doi:10.1111/oik.01248).
- [17] Boyer, N., Réale, D., Marmet, J., Pisanu, B. & Chapuis, J.-L. 2010 Personality, space use and tick load in an introduced population of Siberian chipmunks *Tamias sibiricus*. *J. Anim. Ecol.* **79**, 538-547. (doi:10.1111/j.1365-2656.2010.01659.x).
- [18] Payne, E., Sinn, D.L., Spiegel, O., Leu, S.T., Wohlfeil, C., Godfrey, S.S., Gardner, M. & Sih, A. 2020 Consistent individual differences in ecto-parasitism of a long-lived lizard host. *Oikos* **129**, 1061-1071. (doi:10.1111/oik.06670).
- [19] Martin, L.B., Addison, B., Bean, A.G.D., Buchanan, K.L., Crino, O.L., Eastwood, J.R., Flies, A.S., Hamede, R., Hill, G.E., Klaassen, M., et al. 2019 Extreme competence: keystone hosts of infections. *Trends in Ecology and Evolution* **34**, 303-314. (doi:10.1016/j.tree.2018.12.009).
- [20] Wolff, J.O. 1996 Population fluctuations of mast-eating rodents are correlated with production of acorns. *J. Mammal.* **77**, 850-856. (doi:10.2307/1382690).
- [21] Schaub, E.M. & Ostfeld, R.S. 2002 Modeling the effects of reservoir competence decay and demographic turnover in Lyme disease ecology. *Ecol. Appl.* **12**, 1142-1162. (doi:10.1890/1051-0761(2002)012[1142:MTEORC]2.0.CO;2).

- [22] Hersh, M.H., LaDeau, S.L., Previtalli, M.A. & Ostfeld, R.S. 2014 When is a parasite not a parasite? Effects of larval tick burdens on white-footed mouse survival. *Ecology* **95**, 1360-1369. (doi:10.1890/12-2156.1).
- [23] Voordouw, M.J., Lachish, S. & Dolan, M.C. 2015 The Lyme disease pathogen has no effect on the survival of its rodent reservoir host. *PLOS ONE* **10**, e0118265. (doi:10.1371/journal.pone.0118265).
- [24] Bian, L. 2003 The representation of the environment in the context of individual-based modeling. *Ecol. Model.* **159**, 279-296. (doi:10.1016/S0304-3800(02)00298-3).
- [25] Synes, N.W., Watts, K., Palmer, S.C.F., Bocedi, G., Bartoń, K.A., Osborne, P.E. & Travis, J.M.J. 2015 A multi-species modelling approach to examine the impact of alternative climate change adaptation strategies on range shifting ability in a fragmented landscape. *Ecological Informatics* **30**, 222-229. (doi:10.1016/j.ecoinf.2015.06.004).
- [26] Zuur, A.F., Ieno, E.N. & Smith, G.M. 2007 *Analysing ecological data*. New York, NY, USA, Springer; 672 p.
- [27] Levin, M.L. & Fish, D. 1998 Density-dependent factors regulating feeding success of *Ixodes scapularis* larvae (Acari: Ixodidae). *The Journal of Parasitology* **84**, 36-43. (doi:10.2307/3284526).
- [28] Randolph, S.E. 1979 Population regulation in ticks: the role of acquired resistance in natural and unnatural hosts. *Parasitology* **79**, 141-156. (doi:10.1017/S0031182000052033).
- [29] Kilpatrick, M.A., Dobson, A.D.M., Levi, T., Salkeld, D.J., Swei, A., Ginsberg, H.S., Kjemtrup, A., Padgett, K.A., Jensen, P.M. & Fish, D. 2017 Lyme disease ecology in a changing world: consensus, uncertainty and critical gaps for improving control. *Philosophical Transactions of the Royal Society B: Biological Sciences* **372**, 20160117. (doi:10.1098/rstb.2016.0117).
- [30] Ogden, N.H., Bigras-Poulin, M., O'Callaghan, C.J., Barker, I.K., Lindsay, L.R., Maarouf, A., Smoyer-Tomic, K.E., Waltner-Toews, D. & Charron, D. 2005 A dynamic population model to investigate effects of climate on geographic range and seasonality of the tick *Ixodes scapularis*. *Int. J. Parasitol.* **35**, 375-389. (doi:10.1016/j.ijpara.2004.12.013).
- [31] Hancock, P.A., Brackley, R. & Palmer, S.C.F. 2011 Modelling the effect of temperature variation on the seasonal dynamics of *Ixodes ricinus* tick populations. *Int. J. Parasitol.* **41**, 513-522. (doi:10.1016/j.ijpara.2010.12.012).
- [32] Brunner, J.L. & Ostfeld, R.S. 2008 Multiple causes of variable tick burdens on small-mammal hosts. *Ecology* **89**, 2259-2272. (doi:10.1890/07-0665.1).

- [33] Schmidt, K.A., Ostfeld, R.S. & Schaubert, E.M. 1999 Infestation of *Peromyscus leucopus* and *Tamias striatus* by *Ixodes scapularis* (Acari: Ixodidae) in relation to the abundance of hosts and parasites. *J. Med. Entomol.* **36**, 749-757. (doi:10.1093/jmedent/36.6.749).
- [34] Estrada-Peña, A., Ayllón, N. & De La Fuente, J. 2012 Impact of climate trends on tick-borne pathogen transmission. *Frontiers in Physiology* **3**, 64. (doi:10.3389/fphys.2012.00064).
- [35] Ogden, N.H., Ben Beard, C., Ginsberg, H.S. & Tsao, J.I. 2020 Possible effects of climate change on Ixodid ticks and the pathogens they transmit: predictions and observations. *J. Med. Entomol.* **58**, 1536-1545. (doi:10.1093/jme/tjaa220).
- [36] Hasle, G. 2013 Transport of ixodid ticks and tick-borne pathogens by migratory birds. *Frontiers in Cellular and Infection Microbiology* **3**, 48. (doi:10.3389/fcimb.2013.00048).
- [37] Myrsetrud, A., Heylen, D.J.A., Matthysen, E., Garcia, A.L., Jore, S. & Viljugrein, H. 2019 Lyme neuroborreliosis and bird populations in northern Europe. *Proc. R. Soc. B* **286**, 20190759. (doi:10.1098/rspb.2019.0759).
- [38] Buczek, A.M., Buczek, W., Buczek, A. & Bartosik, K. 2020 The potential role of migratory birds in the rapid spread of ticks and tick-borne pathogens in the changing climatic and environmental conditions in Europe. *International Journal of Environmental Research and Public Health* **17**, 2117. (doi:10.3390/ijerph17062117).
- [39] Schneider, S.C., Parker, C.M., Miller, J.R., Page Fredericks, L. & Allan, B.F. 2015 Assessing the contribution of songbirds to the movement of ticks and *Borrelia burgdorferi* in the Midwestern United States during fall migration. *EcoHealth* **12**, 164-173. (doi:10.1007/s10393-014-0982-3).
- [40] Scott, J.D., Fernando, K., Banerjee, S.N., Durden, L.A., Byrne, S.K., Banerjee, M., Mann, R.B. & Morshed, M.G. 2001 Birds disperse ixodid (Acari: Ixodidae) and *Borrelia burgdorferi*-infected ticks in Canada. *J. Med. Entomol.* **38**, 493-500. (doi:10.1603/0022-2585-38.4.493).
- [41] Ogden, N.H., Lindsay, L.R., Hanincová, K., Barker, I.K., Bigras-Poulin, M., Charron, D.F., Heagy, A., Francis, C.M., O'Callaghan, C.J. & Schwartz, I. 2008 Role of migratory birds in introduction and range expansion of *Ixodes scapularis* ticks and of *Borrelia burgdorferi* and *Anaplasma phagocytophilum* in Canada. *Appl. Environ. Microbiol.* **74**, 1780-1790. (doi:10.1128/AEM.01982-07).
- [42] Altwegg, R., Collingham, Y.C., Erni, B. & Huntley, B. 2013 Density-dependent dispersal and the speed of range expansions. *Divers. Distrib.* **19**, 60-68. (doi:10.1111/j.1472-4642.2012.00943.x).

- [43] Hein, S., Pfenning, B., Hovestadt, T. & Poethke, H.J. 2004 Patch density, movement pattern, and realised dispersal distances in a patch-matrix landscape - a simulation study. *Ecol. Model.* **174**, 411-420. (doi:10.1016/j.ecolmodel.2003.10.005).
- [44] Perkins, S.E., Cattadori, I.M., Tagliapietra, V., Rizzoli, A.P. & Hudson, P.J. 2003 Empirical evidence for key hosts in persistence of a tick-borne disease. *Int. J. Parasitol.* **33**, 909-917. (doi:10.1016/S0020-7519(03)00128-0).
- [45] Sih, A., Bell, A.M., Johnson, J.C. & Ziemba, R.E. 2004 Behavioral syndromes: an integrative overview. *The Quarterly Review of Biology* **79**, 241-277. (doi:10.1086/422893).
- [46] Boon, A.K., Réale, D. & Boutin, S. 2008 Personality, habitat use, and their consequences for survival in North American red squirrels *Tamiasciurus hudsonicus*. *Oikos* **117**, 1321-1328. (doi:10.1111/j.0030-1299.2008.16567.x).
- [47] Cote, J., Fogarty, S., Weinersmith, K., Brodin, T. & Sih, A. 2010 Personality traits and dispersal tendency in the invasive mosquitofish (*Gambusia affinis*). *Proc. R. Soc. B* **277**, 1571-1579. (doi:10.1098/rspb.2009.2128).
- [48] Seyfarth, R.M., Silk, J.B. & Cheney, D.L. 2012 Variation in personality and fitness in wild female baboons. *Proc. Natl. Acad. Sci. U. S. A.* **109**, 16980-16985. (doi:10.1073/pnas.1210780109).
- [49] Nathan, R., Getz, W.M., Revilla, E., Holyoak, M., Kadmon, R., Saltz, D. & Smouse, P.E. 2008 A movement ecology paradigm for unifying organismal movement research. *Proceedings of the National Academy of Sciences* **105**, 19052-19059. (doi:10.1073/pnas.0800375105).